# Normality Calibration in Semi-supervised Graph Anomaly Detection

## Abstract

Graph anomaly detection (GAD) has attracted growing interest for its crucial ability to uncover irregular patterns in broad applications. Semi-supervised GAD, which assumes a subset of annotated normal nodes available during training, is among the most widely explored application settings. However, the normality learned by existing semi-supervised GAD methods is limited to the labeled normal nodes, often inclining to overfitting the given patterns. These can lead to high detection errors, such as high false positives. To overcome this limitation, we propose **GraphNC**, a graph normality calibration framework that leverages both labeled and unlabeled data to calibrate the normality from a teacher model (a pre-trained semi-supervised GAD model) jointly in anomaly score and node representation spaces. GraphNC includes two main components, anomaly score distribution alignment (**ScoreDA**) and perturbation-based normality regularization (**NormReg**). ScoreDA optimizes the anomaly scores of our model by aligning them with the score distribution yielded by the teacher model. Due to accurate scores in most of the normal nodes and part of the anomaly nodes in the teacher model, the score alignment effectively pulls the anomaly scores of the normal and abnormal classes toward the two ends, resulting in more separable anomaly scores. Nevertheless, there are inaccurate scores from the teacher model. To mitigate the misleading by these scores, NormReg is designed to regularize the graph normality in the representation space, making the representations of normal nodes more compact by minimizing a perturbation-guided consistency loss solely on the labeled nodes. Through comprehensive experiments on six benchmark datasets, we show that, by jointly optimizing these two components, GraphNC can 1) consistently and substantially enhance the GAD performance of teacher models from different types of GAD methods and 2) achieve new state-of-the-art semi-supervised GAD performance.

## 1 Introduction

Graph anomaly detection (GAD), aiming to identify irregular nodes in a graph, has gained increasing attention due to its critical role in real applications such as detection of spams in social networks and frauds in financial networks (Pang et al., 2021a; Ma et al., 2021; Qiao et al., 2025). Semi-supervised GAD, which leverages a subset of labeled normal nodes to learn normal patterns, has gained significant attention in real-world applications (Qiao et al., 2024). This GAD setting is practical in that normal nodes often account for the majority in a graph, and thus, it is significantly less costly to obtain compared to abnormal samples that are scarce and their occurrence typically involves high financial/reputation loss. However, compared to unsupervised GAD that is more popular, less studies have been done on semi-supervised GAD. Furthermore, existing semi-supervised GAD methods are primarily built on the limited labeled normal nodes, preventing them from learning the exact normal pattern, thus inclining to overfit the annotated normality (Ding et al., 2019; Wang et al., 2021; Ding et al., 2021; Fan et al., 2020; Qiao et al., 2024). As a result, these methods suffer from high detection errors—*e.g.*, normal nodes that are dissimilar to the labeled normal nodes are detected as anomalies (false positives) and vice versa (false negatives), especially the former one—due to the large overlapped anomaly scores for the normal and abnormal nodes. These issues can be observed in the results of a recent state-of-the-art (SOTA) method GGAD (Qiao et al., 2024) in Figs. 1 (a)

and (b). The same phenomenon can also be observed in other semi-supervised methods like data reconstruction and one-class classification-based methods (see App. C).

To address these issues, we propose **GraphNC**, a novel graph normality calibration framework that exploits both labeled and unlabeled data to calibrate the normality learned from a teacher model (*i.e.*, a pre-trained semi-supervised GAD model) jointly in anomaly score and representation spaces.

GraphNC includes two main components, namely anomaly score distribution alignment (**ScoreDA**) and perturbation-based normality regularization (**NormReg**). ScoreDA optimizes the anomaly scores of our model by aligning them with the score distribution yielded by the teacher model. Due to accurate scores in most of the normal nodes and part of the anomaly nodes in the teacher model, the score alignment effectively pulls the anomaly scores of the normal and abnormal classes toward the two ends, resulting in more separable anomaly scores, as shown in Fig.1 (d). This helps reduce false positive and false negative rates.

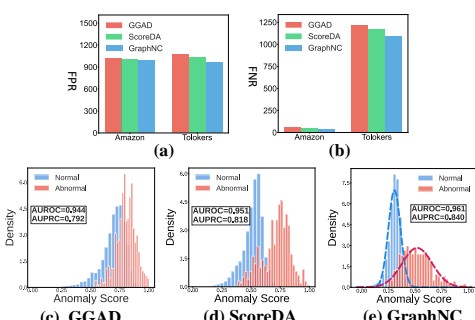

(a)        (b)

(c) GGAD     (d) ScoreDA     (e) GraphNC

Figure 1: **(a)** False positive rate and **(b)** false negative rate results on Amazon (Dou et al., 2020) and Tolokers (McAuley et al., 2015). **(c)**, **(d)**, and **(e)** show the score distributions of normal and abnormal nodes for GGAD, ScoreDA, and ScoreDA+NormReg (*i.e.*, GraphNC) on Amazon, where GGAD is used as a teacher model in both ScoreDA and GraphNC.

However, due to their inherent limitation mentioned above, the teacher models inevitably produce some inaccurate anomaly scores, which may mislead the score optimization in ScoreDA. To mitigate this issue, NormReg is devised to regularize the graph normality in the representation space, enabling the student model to refine itself by minimizing a perturbation-guided consistency loss solely on the labeled nodes. This normality calibration allows our model to learn more compact normal representations, which helps pull the anomaly scores of many normal nodes closer to each other and toward the lower end of the anomaly score distribution, as shown in Fig. 1 (e).

By jointly optimizing these two components, GraphNC learns more generalized normality representations and more discriminative anomaly scores. Besides, GraphNC is a generic framework where different pre-trained teacher models can be plugged into, and it can obtain better GAD performance if the teacher model is stronger. In summary, our contributions are as follows

- We introduce GraphNC, a novel graph normality calibration framework that leverages both labeled normal nodes and unlabeled data to calibrate normality learning from the teacher model in both the score and feature spaces for semi-supervised GAD .

- In GraphNC, we introduce two new components: anomaly score distribution alignment (ScoreDA) and perturbation-based normality regularization (NormReg). ScoreDA aligns the scores of our model with those of the pre-trained teacher model, which helps push the anomaly scores of the two classes toward opposite ends. Meanwhile, NormReg mitigates the impact of inaccurate teacher scores by enforcing representation compactness among normal nodes during alignment.

- GraphNC is a flexible framework where different teacher models can be plugged and played, achieving consistently enhanced GAD improvement across three types of teacher models, including data reconstruction-based, one-class-based, and anomaly-generation-based approaches. Further, GraphNC gets better performance when the teacher model is stronger. This is verfied by our comprehensive experiments on six benchmark datasets.

## 2  RELATED WORK

**Graph Anomaly Detection.** Existing GAD methods can be broadly categorized into unsupervised, semi-supervised, and fully supervised approaches (Ma et al., 2021; Qiao et al., 2025). Unsupervised methods typically assume that no labeled nodes are known, and they learn normal patterns through

some proxy task designs such as reconstruction, one-class classification, and adversarial learning (Ding et al., 2019; Wang et al., 2021; Ding et al., 2021). Although this setting is widely used in anomaly detection, they overlook the abundance of normal samples, where normal nodes are less costly to obtain than the abnormal nodes. The fully supervised methods, on the other hand, rely on labeled normal and abnormal nodes and formulate the task as an imbalanced classification problem (Liu et al., 2021a; Tang et al., 2022; Gao et al., 2023a;b; Chen et al., 2024; Liu et al., 2025). These supervised methods rely on large labeled data, limiting their applicability in real-world scenarios, especially those where anomalies are very infrequent and very costly to collect; they may overfit labeled anomalies, failing to generalize to unseen anomalies (Wang et al., 2025). Semi-supervised methods are generally more practical for real-world applications, as only a subset of labeled normal nodes is required (Qiao et al., 2024; 2025). Only limited work has been done in this line. GGAD (Qiao et al., 2024) is a recent work that utilizes the labeled normal nodes to generate synthetic outliers that mimic real anomalies, and then trains a binary classifier on this data. It also shows that popular unsupervised methods, such as reconstruction, one-class classification, or adversarial learning-based methods, can be adapted to the semi-supervised settings by minimizing their loss on the normal data only.

**Semi-supervised Learning.** Semi-supervised learning aim to leverage a small amount of labeled data together with a large amount of unlabeled data to train models, aiming to achieve high performance while reducing the cost of data annotation (Yang et al., 2022; Chen et al., 2022b). Representative semi-supervised learning approaches include pseudo-labeling and consistency-based methods. The pseudo-labeling with confidence thresholding is highly successful and widely-adopted, which is typically built on the smoothness assumption that neighboring data points in the feature space tend to exhibit similar labels (Grandvalet & Bengio, 2004; Chen et al., 2023). However, such a mechanism requires a quantity–quality trade-off, which undermines the learning process, since incorporating more unlabeled data helps increase data coverage but often introduces noisy or unreliable pseudo-labels (Pham et al., 2021; Kage et al., 2024). The consistency-based model enforces prediction consistency under stochastic noise or data augmentation (Laine & Aila, 2016). Recent studies such as RankMatch (Mai et al., 2024), InterLUDE (Huang et al., 2024), and SCHOOL (Mo et al., 2024), have been proposed in different data modalities, including visual images, text, and graphs, aiming to encourage stable predictions across views, thereby enhancing generalization and reducing reliance on labeled data (Gui et al., 2024). However, they are mainly focused on classification tasks, different from our anomaly detection task, which has only one-class labels.

Many semi-supervised anomaly detection models that are trained on normal data have been proposed for anomaly detection tasks, but they are focused on visual data (Wu et al., 2024; Cao et al., 2024), failing to capture the complex structural information in the graph. There are also methods that aim to leverage small labeled anomaly data and large unlabeled data, such as DevNet (Pang et al., 2019), Deep SAD (Ruff et al., 2019), DPLAN (Pang et al., 2021b), PReNet (Pang et al., 2023), and RoSAS (Xu et al., 2023), which explore a different problem setting from ours.

## 3 PROBLEM STATEMENT

**Notations.** Given an attributed graph $\mathcal{G} = (\mathcal{V}, \mathcal{E}, \mathbf{X})$, where $\mathcal{V}$ denotes the node set with $v_i \in \mathcal{V}$ and $|\mathcal{V}| = N$ representing the total number of the node, $\mathcal{E}$ denotes the edge set, and $\mathbf{X} = [\mathbf{x}_1, \mathbf{x}_2, \ldots, \mathbf{x}_N] \in \mathbb{R}^{N \times M}$ is a set of node attributes. Each node $v_i$ has a $M$-dimensional attribute $\mathbf{x}_i \in \mathbb{R}^M$. The topological structure of $\mathcal{G}$ is represented by an adjacency matrix $\mathbf{A} \in \mathbb{R}^{N \times N}$.

**Semi-supervised GAD.** Let $\mathcal{V}_a, \mathcal{V}_n$ be two disjoint subsets of $\mathcal{V}$, where $\mathcal{V}_a$ represents abnormal node set and $\mathcal{V}_n$ represents normal node set, and typically the number of normal nodes is significantly greater than the abnormal nodes, $i.e.$, $|\mathcal{V}_n| \gg |\mathcal{V}_a|$, then the goal of semi-supervised GAD is to learn the mapping function $f \to \mathbb{R}$, such that $f(v) < f(v')$, where $\forall v \in \mathcal{V}_n, v' \in \mathcal{V}_a$ , given a set of labeled normal nodes $\mathcal{V}_l \subset \mathcal{V}_n$, with $|\mathcal{V}_l| = R$, and no access to any labels of the abnormal nodes. $\mathcal{V}_u = \mathcal{V}/\mathcal{V}_l$ is the set of the unlabeled nodes and used as test data.

**Graph Neural Networks for Representation Learning.** Graph Neural Networks (GNNs) are typically employed to learn the representation of each node due to their strong representation ability. This can be formulated as follows

$$\mathbf{H}^{(\ell)} = \text{GNN}\left(\mathbf{A}, \mathbf{H}^{(\ell-1)}; \mathbf{W}^{(\ell)}\right) \tag{1}$$

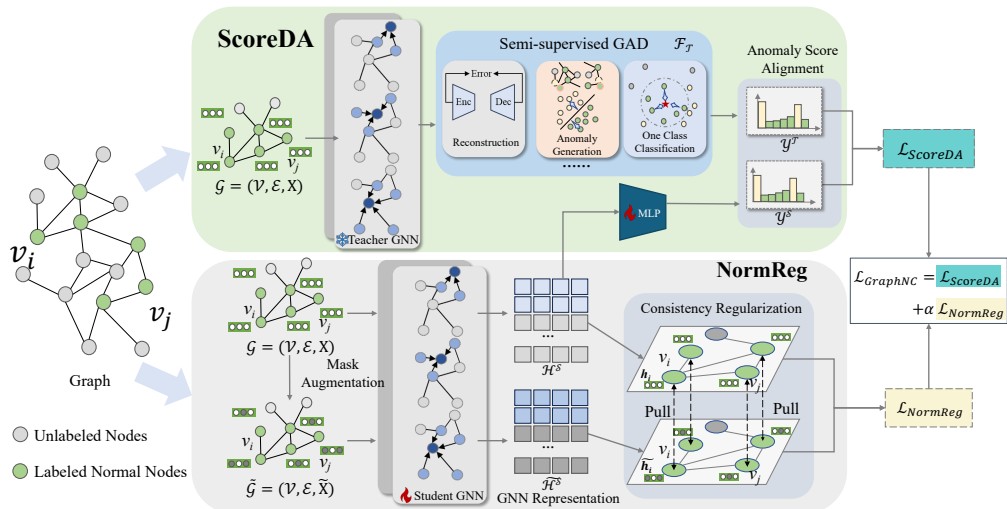

Figure 2: Overview of GraphNC. The input graph consists of a small labeled normal node set and a large unlabeled node set. GraphNC is based on a teacher-student network framework. ScoreDA aims to align the anomaly scores of the student network with the anomaly scores yielded by an existing semi-supervised GAD method to calibrate the normality in the score space. NormReg, on the other hand, is introduced to utilize a consistency regularization loss function in representation space to learn more compact representations of normal nodes, thereby reducing the negative impact of inaccurate anomaly scores that may be produced by the teacher model. The teacher network is pre-trained first, and it is frozen afterward, with only the student being trained.

where $\mathbf{W}^{(\ell)}$ are the parameters, which are updated during training, and $\mathbf{H}^{(\ell)} = \{\mathbf{h}_1, \mathbf{h}_2, ..., \mathbf{h}_N\}$ denote the $l$-th (final) layer embeddings, which are the representation of all nodes at layer $l$. In this paper, our student model adopts a 2-layer GNN as the backbone for representation learning, while the teacher model uses the recommended GNN architecture in its original work.

## 4 METHODOLOGY

### 4.1 OVERVIEW OF THE PROPOSED GRAPHNC

The overview of the proposed GraphNC is shown in the Fig. 2. Given a pre-trained semi-supervised GAD model, it includes two steps during training. (1) Given a graph with only a set of labeled normal nodes, ScoreDA aligns the scores of our student model with the anomaly score distribution produced by the pre-trained detector as the teacher model, thereby achieving normality calibration in the score space. (2) NormReg subsequently complements score alignment by mitigating noise in the teacher model's anomaly scores through a consistency constraint in the representation space on the labeled normal nodes, encouraging them to learn more compact normality representations. During inference, the output scores from the student model are used to derive the anomaly scores.

### 4.2 ANOMALY SCORE DISTRIBUTION ALIGNMENT (SCOREDA)

The simplistic normal pattern extraction in existing semi-supervised GAD models, trained on only a limited set of labeled normal samples, often is prone to overfit the patterns in the labeled data, resulting in high detection errors, such as elevated false positive rates, *i.e.*, many normal samples are misclassify as anomalies. We leverage ScoreDA to align the output scores of our model with the score distribution produced by the teacher model. As the anomaly score distribution of the pre-trained teacher model can correctly characterize most nodes, including most normal nodes and part of the abnormal nodes in the score space, by aligning with this teacher scores, the student model is

enforced to push scores for the normal and abnormal classes toward the two opposite ends, resulting in more separable anomaly scores. This effectively reduces the misclassification of normal samples, leading to a lower false positive rate, while also yielding a decrease in the false negative rate.

Formally, let $\mathcal{F}_{\mathcal{T}}$ be the pre-trained teacher model, $\mathcal{Y}^{\mathcal{T}} = \mathcal{F}_{\mathcal{T}}(\mathbf{X}, \mathcal{G}; \Theta)$ be the output scores for the entire node set, where $\Theta$ are the trainable parameters, and $\mathcal{Y}^{\mathcal{T}} = \{y_1^{\mathcal{T}}, y_2^{\mathcal{T}}, ..., y_N^{\mathcal{T}}\}$ be the anomaly score distribution of samples, then we leverage a student model $\mathcal{F}_S$ with parameter $\Phi$ which is implemented as the combination of GNN and MLP layers (*i.e.*, $\mathcal{F}_{GNN}$ with parameters $\Omega$ and $\mathcal{F}_{MLP}$ with parameters $\phi$ respectively) to align the score set $\mathcal{Y}^{\mathcal{T}}$. For the student model, let $\mathcal{H}^S$ be the output of the GNN student model, for a node $v_i$, the representation is then denoted by $\mathbf{h}_i^{\mathcal{S}} = \mathcal{F}_{GNN}(\mathbf{x}_i, \mathcal{G}; \Omega)$. On top of that, we apply a MLP layer to obtain the predicted score, $y_i^{\mathcal{S}} = \mathcal{F}_{MLP}(\mathbf{h}_i^{\mathcal{S}}; \phi)$. Finally, we employ the MSE loss to minimize the discrepancy between the anomaly score from the student model $\mathcal{Y}^{\mathcal{S}}$ and $\mathcal{Y}^{\mathcal{T}}$ yielded by the teacher model, which is formulated as follows:

$$\mathcal{L}_{ScoreDA} = \frac{1}{|\mathcal{V}|} \sum_{v_i \in \mathcal{V}} \|y_i^{\mathcal{S}} - y_i^{\mathcal{T}}\|_2^2, \tag{2}$$

where $|\mathcal{V}|$ is the number of all nodes, including both labeled and unlabeled nodes. By minimizing the score discrepancy between the teacher and the student, the accurate scores in $\mathcal{Y}^{\mathcal{T}}$ helps enforce the clusters of the anomaly scores in the normal and abnormal classes in the two opposite ends, resulting in more separable anomaly scores.

### 4.3 Perturbation-Guided Normality Regularization (NormReg)

Due to the inherent limitations in normality characterization and the reliance on a limited set of normal samples, teacher models inevitably produce some inaccurate scores. Solely applying score alignment can enforce the fitting of the student model to these inaccurate scores, thereby misleading the optimization of the student model. To address this challenge, we further introduce NormReg to calibrate the learned normality in the representation space based on solely the labeled normal nodes, enabling the student to not only learn from the teacher but also self-refine.

To this end, we utilize an node attribute-based masking mechanism that randomly masks a proportion $\omega$ of the attributes on labeled normal nodes to learn consistent representations for these nodes. This masking creates an augmented graph $\tilde{\mathcal{G}}$, which contains many variations of the labeled normal nodes, simulating diverse normal patterns that may be different from the ones derived directly from the labeled nodes. Thus, enforcing consistency over the representations of these augmented nodes and the labeled normal nodes helps cluster the normality from both the labeled nodes and their augmented versions closer. Formally, let $\mathcal{H}_i^{\mathcal{S}}$ and $\tilde{\mathcal{H}}_i^{\mathcal{S}}$ be the original and augmented node representations in the student model respectively, NormReg then minimizes the discrepancy between the representations of the original node and its augmented counterpart by optimizing the following loss function $\mathcal{L}_{NormReg}$:

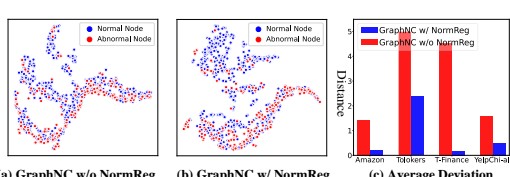

(a) GraphNC w/o NormReg    (b) GraphNC w/ NormReg    (c) **Average Deviation**

Figure 3: (a) and (b) provide t-SNE visualization of the node representations for GraphNC with/without using NormReg. (c) The average deviation of the normal class on Tolokers.

$$\mathcal{L}_{NormReg} = \frac{1}{|\mathcal{V}_l|} \sum_{v_i \in \mathcal{V}_l} \left\| \mathbf{h}_i^{\mathcal{S}} - \tilde{\mathbf{h}}_i^{\mathcal{S}} \right\|_2^2, \tag{3}$$

where $\mathcal{V}_l$ are the labeled normal nodes, and $\mathbf{h}_i^{\mathcal{S}} \in \mathcal{H}_i^{\mathcal{S}}$ and $\tilde{\mathbf{h}}_i^{\mathcal{S}} \in \tilde{\mathcal{H}}_i^{\mathcal{S}}$ are the original and augmented representations for node $v_i$ respectively. As shown in Figs. 3 (a) and (b), by adding $\mathcal{L}_{NormReg}$, the representation distributions of the normal nodes obtained by GraphNC are more compact compared to GraphNC without using $\mathcal{L}_{NormReg}$. This demonstrates that the regularization effectively mitigates the negative impact of inaccurate anomaly scores from the teacher model. To provide more evidence, we also compute average deviation of normal nodes to the normal class prototype on four datasets, and it is clear that the deviation is significantly reduced after applying NormReg, as illustrated in Fig. 3 (c). Beyond the empirical demonstrations, our theoretical analysis also show that NormReg can well complement ScoreDA in lowering the detection errors.

**Theorem 1.** *Let $\sigma_{\mathcal{T}}^2$ and $\sigma_{\mathcal{S}}^2$ denote the variance of the anomaly scores yielded by the teacher model and the student model in the normal class, respectively, then under our GraphNC framework, the student model achieves shrinking score variance (i.e., $\sigma_{\mathcal{S}}^2 < \sigma_{\mathcal{T}}^2$) by minimizing $\mathcal{L}_{NormReg}$ while applying $\mathcal{L}_{ScoreDA}$, thereby reducing False Positive Rate (FPR) and False Negative Rate (FNR).*

The proof can be found in App. D. According to Theorem 1, minimizing $\mathcal{L}_{NormReg}$ complements $\mathcal{L}_{ScoreDA}$ in reducing the intra-class score variance of normal classes in the student model, which can also be observed when comparing the scores of the normal class in Fig. 1(e) to that in Fig. 1(d), lowering both FPR and FNR, especially the FPR.

### 4.4 TRAINING AND INFERENCE

**Training.** During training, the student model is optimized by aligning its score distribution with that of the teacher model, while also minimizing the consistency regularization. To be specific, the total loss function $\mathcal{L}_{GraphNC}$ is formulated as a combination of the $\mathcal{L}_{ScoreDA}$ and $\mathcal{L}_{NormReg}$:

$$\mathcal{L}_{GraphNC} = \mathcal{L}_{ScoreDA} + \alpha \mathcal{L}_{NormReg}, \tag{4}$$

where $\alpha$ is a hyper-parameter that adjusts the influence of normality consistency regularization in optimization.

**Inference.** During inference, since the student model $\mathcal{F}_{\mathcal{S}}$ calibrates the normality of the teacher and yields more accurate scores, its output scores can be directly employed as an anomaly score, which is defined as:

$$S(v_i) = \mathcal{F}_{\mathcal{S}}(v_i, \mathbf{x}_i, \mathcal{G}, \Phi^*), \tag{5}$$

where $\Phi^* = \{\Omega^*, \phi^*\}$ is the optimized parameter set of the student model.

## 5 EXPERIMENTS

**Datasets.** We evaluate the effectiveness of the GraphNC across six real-world GAD datasets drawn from diverse domains, including social networks: Reddit (Kumar et al., 2019)), online shopping co-review networks: Amazon (Dou et al., 2020) and YelpChi (Dou et al., 2020), a co-purchase network: Photo (Shchur et al., 2018), a collaboration network: Tolokers (McAuley et al., 2015), and financial networks: T-Finance (Tang et al., 2022). Additional description and statistical results are provided in App. A. Note that some unsupervised methods, such as CoLA (Liu et al., 2021b), GRADATE (Duan et al., 2023), and HUGE (Pan et al., 2025), cannot be adapted to semi-supervised setting due to their design more relying on fully unlabeled data, which are excluded from our comparison.

**Competing Methods.** The competing methods include reconstruction-based methods: DOMI-NANT (Ding et al., 2019), GADNR (Roy et al., 2024), ADA-GAD (He et al., 2024) and Anomaly-DAE (Fan et al., 2020), one-class classification: OCGNN (Wang et al., 2021), adversarial learning based methods: AEGIS (Ding et al., 2021) and GAAN (Chen et al., 2020), affinity maximization-based methods: TAM (Qiao & Pang, 2023), and anomaly generation-based method: GGAD (Qiao et al., 2024). A more detailed description of competing methods can be found in the App. B. Except GGAD that is specifically designed for the semi-supervised setting, the competing methods are primarily unsupervised methods, which are adapted to our semi-supervised setting following (Qiao et al., 2024): reconstruction is performed using only the normal nodes, the one-class center calculation is based on only the normal nodes, the adversarial discriminator is trained with the normal nodes, and the affinity maximization is also applied on the normal nodes solely.

**Evaluation Metric.** Following (Liu et al., 2021b; Qiao et al., 2024), two widely-used metrics, AUROC and AUPRC, are used to evaluate the performance of all methods. For both metrics, a higher value denotes a better performance. Moreover, for each method, we report the average performance after 5 independent runs with different random seeds.

**Implementation Details.** Our GraphNC model is implemented in Pytorch 1.10.0 with Python 3.8, and all the experiments are performed using GeForce RTX 3090 (24GB). GraphNC is optimized using the Adam optimizer (Kinga et al., 2015) with a learning rate of $5e-3$ for Photo and Reddit that have relatively high attribute dimensions, and $5e-4$ for Amazon, T-Finance, YelpChi, and Tolokers that have relatively low attribute dimensions, to avoid overfitting. The pre-trained teacher is set to GGAD (Qiao et al., 2024) by default, the experimental results of other teacher models can

Table 1: AUROC and AUPRC results on six real-world GAD datasets. For each dataset, the best performance per column within each metric is boldfaced, with the second-best underlined. "Avg" denotes the average performance of each method.

| Metric | Method | Dataset | | | | | | |
|--------|--------|---------|-----------|--------|---------|----------|--------|--------|
| | | Amazon | T-Finance | Reddit | YelpChi | Tolokers | Photo | Avg. |
| AUROC | DOMINANT | 0.8867 | 0.6167 | 0.5194 | 0.6314 | 0.5121 | 0.5314 | 0.6162 |
| | AnomalyDAE | 0.9171 | 0.6027 | 0.5280 | 0.6161 | 0.6047 | 0.5272 | 0.6326 |
| | OCGNN | 0.8810 | 0.5742 | 0.5622 | 0.6155 | 0.4803 | 0.6461 | 0.6265 |
| | AEGIS | 0.7593 | 0.6728 | 0.5605 | 0.6022 | 0.4451 | 0.5936 | 0.6055 |
| | GAAN | 0.6531 | 0.3636 | 0.5349 | 0.5605 | 0.3785 | 0.4355 | 0.4876 |
| | TAM | 0.8405 | 0.5923 | 0.5829 | 0.6075 | 0.4847 | 0.6013 | 0.6182 |
| | GADNR | 0.6436 | 0.5943 | 0.5978 | 0.4554 | 0.6406 | 0.6158 | 0.5913 |
| | ADA-GAD | 0.2034 | 0.4416 | 0.5765 | 0.4400 | 0.5369 | 0.4067 | 0.4342 |
| | GGAD | 0.9443 | 0.8228 | 0.6354 | 0.6180 | 0.5340 | 0.6476 | 0.7003 |
| | GraphNC | **0.9613** | **0.8340** | **0.6420** | **0.6630** | **0.6505** | **0.7693** | **0.7533** |
| AUPRC | DOMINANT | 0.7289 | 0.0542 | 0.0414 | 0.2237 | 0.2217 | 0.1283 | 0.2330 |
| | AnomalyDAE | 0.7748 | 0.0538 | 0.0362 | 0.2064 | 0.2697 | 0.1177 | 0.2431 |
| | OCGNN | 0.6895 | 0.0492 | 0.0400 | 0.1937 | 0.2138 | 0.1501 | 0.2227 |
| | AEGIS | 0.2616 | 0.0685 | 0.0441 | 0.2192 | 0.1943 | 0.1110 | 0.1497 |
| | GAAN | 0.0856 | 0.0324 | 0.0362 | 0.1724 | 0.1693 | 0.0768 | 0.0945 |
| | TAM | 0.5183 | 0.0551 | 0.0446 | 0.2184 | 0.2178 | 0.1087 | 0.1938 |
| | GADNR | 0.0866 | 0.0523 | 0.0456 | 0.1324 | **0.3511** | 0.1430 | 0.1352 |
| | ADA-GAD | 0.0388 | 0.0375 | 0.0417 | 0.1295 | 0.2566 | 0.0723 | 0.0961 |
| | GGAD | 0.7922 | 0.1825 | 0.0520 | 0.2261 | 0.2502 | 0.1442 | 0.2745 |
| | GraphNC | **0.8403** | **0.3667** | **0.0560** | **0.2389** | 0.3082 | **0.3561** | **0.3610** |

be found in Sec. 5.2. $\alpha$ that controls the importance of NormReg is set to $0.01$ and the mask ratio for the augment mechanism is set $0.30$ by default.

## 5.1 MAIN EXPERIMENTS

The main comparison results are shown in Table 11, where all models use 15% labeled normal nodes during training. From the results, we find that GGAD surpasses reconstruction-based, one-class classification, and adversarial learning approaches, suggesting that the classifier based on anomaly generation is particularly effective for learning the normality in semi-supervised GAD. We observe that GraphNC consistently outperforms all the semi-supervised methods on the six datasets, having an average 7.1% AUROC and 31.5% AUPRC improvement over the best competing method GGAD. These experimental results show that despite building upon GGAD, GraphNC consistently outperforms GGAD, indicating that the two components in GraphNC can effectively calibrate the normality learned from GGAD by aligning its scores with the output of GGAD, while at the same time refining itself through representation regularization to mitigate inaccurate anomaly scores from GGAD. Besides, although the reconstruction-based, one-class classification-based, and affinity maximized methods achieve competitive performance on Amazon, T-Finance, and YelpChi, they still largely underperform GraphNC. This is primarily because their modeling of normal patterns is less effective than that of the anomaly-generation-based method GGAD.

## 5.2 GRAPHNC ENABLED THREE TEACHER MODELS

To demonstrate that GraphNC is flexible and can be plugged into various teacher models, in addition to the default teacher model GGAD, we further evaluate the performance GraphNC when plugging the reconstruction-based method DOMINANT and the one-class classification-based method OCGNN as a pre-trained teacher model into our model. The AUROC and AUPRC results are shown in Table 3. It is clear from the results that, similar to the results using GGAD as the teacher model, GraphNC can substantially and consistently improve the performance of DOMINANT and OCGNN in terms of both AUROC and AUPRC. In particular, on Tolokers and Photo, both teacher models struggle to deliver strong performance, whereas applying GraphNC leads to substantial gains in both AUROC and AUPRC, resulting in a 36.2% and 26.3% AUROC improvement on Tolokers for DOMINANT and OCGNN, respectively. The main reason is that aligning the score distribution from the

Table 2: Results of GraphNC using different teacher models. ('*') denotes the teacher model used.

| Metric | Method | Dataset | | | | | | |
| --- | --- | --- | --- | --- | --- | --- | --- | --- |
| | | Amazon | T-Finance | Reddit | YelpChi | Tolokers | Photo | Avg. |
| AUROC | DOMINANT | 0.8867 | 0.6167 | 0.5194 | 0.6314 | 0.5121 | 0.5314 | 0.6162 |
| | GraphNC (DOMINANT) | **0.8936** | **0.7985** | **0.5260** | **0.6543** | **0.6977** | **0.6829** | **0.7088** |
| | OCGNN | 0.8810 | 0.5742 | 0.5622 | 0.6155 | 0.4803 | 0.6461 | 0.6265 |
| | GraphNC (OCGNN) | **0.9255** | **0.6819** | **0.6093** | **0.6447** | **0.6069** | **0.7068** | **0.6956** |
| | GGAD | 0.9443 | 0.8228 | 0.6354 | 0.6180 | 0.5340 | 0.6476 | 0.7003 |
| | GraphNC (GGAD) | **0.9613** | **0.8340** | **0.6420** | **0.6630** | **0.6505** | **0.7693** | **0.7533** |
| AUPRC | DOMINANT | 0.7289 | 0.0542 | **0.0414** | 0.2237 | 0.2217 | 0.1283 | 0.2330 |
| | GraphNC (DOMINANT) | **0.7290** | **0.1703** | **0.0414** | **0.2375** | **0.3465** | **0.1526** | **0.2795** |
| | OCGNN | 0.6895 | 0.0492 | 0.0400 | 0.1937 | 0.2138 | 0.1501 | 0.2227 |
| | GraphNC (OCGNN) | **0.7372** | **0.0748** | **0.0512** | **0.2116** | **0.2814** | **0.1823** | **0.2564** |
| | GGAD | 0.7922 | 0.1825 | 0.0520 | 0.2261 | 0.2502 | 0.1442 | 0.2745 |
| | GraphNC (GGAD) | **0.8403** | **0.3667** | **0.0560** | **0.2389** | **0.3082** | **0.3561** | **0.3610** |

Table 3: AUROC and AUPRC results comparison of GraphNC and its five variants.

| Metric | Method | Amazon | T-Finance | Reddit | YelpChi | Tolokers | Photo | Avg. |
| --- | --- | --- | --- | --- | --- | --- | --- | --- |
| AUROC | OT | 0.9443 | 0.8228 | 0.6354 | 0.6180 | 0.5340 | 0.6476 | 0.7003 |
| | OT+NormReg | 0.8956 | **0.8455** | 0.6317 | 0.6050 | 0.5843 | 0.6731 | 0.7058 |
| | OT+NormReg-Finetune | 0.9060 | 0.8008 | 0.6089 | 0.5729 | 0.5396 | 0.6307 | 0.6764 |
| | OT+ScoreDA | 0.9511 | 0.8300 | 0.6364 | 0.6263 | 0.5811 | 0.7397 | 0.7274 |
| | OT+ScoreDA+NormReg* | 0.9452 | 0.8278 | 0.6212 | 0.6271 | 0.6492 | 0.7433 | 0.7356 |
| | OT+ScoreDA+NormReg | **0.9613** | 0.8340 | **0.6420** | **0.6630** | **0.6505** | **0.7693** | **0.7533** |
| AUPRC | OT | 0.7922 | 0.1825 | 0.0520 | 0.2261 | 0.2502 | 0.1442 | 0.2745 |
| | OT+NormReg | 0.6568 | 0.2413 | 0.0520 | 0.2108 | 0.2688 | 0.1649 | 0.2657 |
| | OT+NormReg-Finetune | 0.7008 | 0.1406 | 0.0454 | 0.1820 | 0.2465 | 0.1310 | 0.2410 |
| | OT+ScoreDA | 0.8189 | 0.2566 | 0.0530 | 0.2111 | 0.2567 | 0.2969 | 0.3155 |
| | OT+ScoreDA+NormReg* | 0.8180 | 0.1971 | 0.0523 | 0.1866 | 0.3034 | 0.2697 | 0.3045 |
| | OT+ScoreDA+NormReg | **0.8403** | **0.3667** | **0.0560** | **0.2389** | **0.3082** | **0.3561** | **0.3610** |

pre-trained teacher model in GraphNC enables the student to calibrate the coarse normality learned by the teacher. This is because that GraphNC can not only inherit the strengths of the teacher but also correct its inaccuracies and refines normality on its own, resulting in consistent performance improvements. Importantly, given a better teacher model, GraphNC can obtain better GAD performance, *e.g.*, GraphNC (GGAD) vs. GraphNC (DOMINANT) and GraphNC (OCGNN).

## 5.3 ABLATION STUDY

In this section, we perform an ablation study to evaluate the contribution of each component in GraphNC. To this end, several variants of GraphNC are introduced. (1) Only Teacher (**OT**) directly uses the pre-trained semi-supervised teacher model to derive the anomaly score. (2) **OT+NormReg** incorporates NormReg into the pre-training of the teacher model. (3) **OT+NormReg+Finetune** leverages NormReg to fine-tune the pre-trained teacher. (4) **OT+ScoreDA** includes the student model and adds ScoreDA on top of OT (*i.e.*, optimizing GraphNC using ScoreDA only). (5) **OT+ScoreDA+NormReg\*** applies NormReg on top of OT+ScoreDA but it applies the NormReg loss on all nodes instead of the labeled normal nodes only. **OT+ScoreDA+NormReg** is the default full model using both ScoreDA and NormReg. The experimental results are shown in Table 3. We observe that OT+ScoreDA enhances the performance of OT, achieving the second-best results on four datasets in terms of AUPRC. This demonstrates that the ScoreDA component can effectively calibrate the normality learned from the teacher, ensuring better separation between normal and abnormal classes. However, OT+ScoreDA underperforms the full model (the last row) on all datasets, highlighting that the negative impact of inaccurate anomaly scores from the teacher model can be effectively mitigated by the NormReg component. On the other hand, applying NormReg directly to the teacher via either joint optimization (OT+NormReg) or finetuning (OT+NormReg+Finetune) can improve OT to some extent. This justifies the effectiveness of NormReg from a different perspective, but NormReg works best when working in the student model and combining with ScoreDA due to

its joint effects in minimizing the detection errors discussed in our theoretical analysis. Besides, we also conduct the ablation study on other teacher models to further analyze the effectiveness of each module in GraphNC. The results can be found in the App. C.2

### 5.4 HYPERPARAMETER SENSITIVITY ANALYSIS

We evaluate the sensitivity of GraphNC w.r.t. the loss moderator $\alpha$, mask ratio $\omega$, and training size $R$. A detailed analysis on time complexity can be found App. E.1.

**Performance w.r.t. the loss moderator $\alpha$.** As shown in Figs. 4 (a-b), our model GraphNC remains generally stable as the consistency regularization weight $\alpha$ varies. However, on Tolokers and YelpChi, the performance slightly declines as $\alpha$ increases. This is mainly because excessive emphasis on the consistency may lead to an overly compressed representation space, which may in turn weaken the representation discriminability.

**Performance w.r.t. the mask ratio $\omega$.** We evaluate GraphNC under different masking ratios. As shown in Figs. 4 (c-d), we observe that increasing the masking ratio $\omega$ generally enhances the performance of GraphNC on Tolokers and T-Finance, while leading to performance degradation on Photo and YelpChi. This suggests that different datasets benefit from different levels of augmentation, depending on the variation in the underlying normality. For consistency, we set $\omega$ to $0.3$ across all datasets.

**Performance w.r.t. the training size $R$.** To explore the impact of training size $R$, we compare GraphNC with four semi-supervised GAD methods using varying numbers of training normal nodes, as shown in Figs.4 (e-f). Similar with other competing methods, with the increase of training size $R$, our model GraphNC generally performs better and it maintains the superiority over other competing methods, demonstrating the effectiveness of GraphNC using different scales of labeled training data.

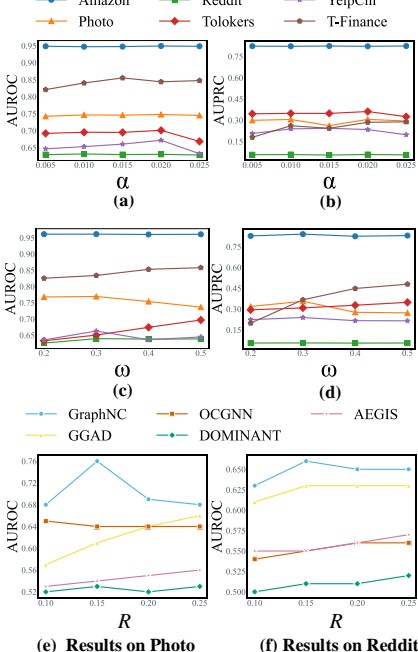

Figure 4: (a-d) AUROC and AURPC results w.r.t $\alpha$ and $\omega$. (e-f) AUROC results w.r.t $R$ on Photo and Reddit.

## 6 CONCLUSION

In this paper, we propose GraphNC, a generic graph normality calibration framework designed to enhance existing semi-supervised GAD methods, achieving substantial and consistent performance improvements across different types of existing approaches. GraphNC is composed of two key components: ScoreDA and NormReg. The ScoreDA module aligns the anomaly scores with the anomaly score distribution produced by the teacher model, effectively pushing the anomaly scores for the normal and abnormal classes toward opposite ends. This alignment ensures a clearer separation between normal and abnormal nodes in the score space, thereby leading to a lower FPR and FNR. The NormReg module is introduced to reduce the impact of inaccurate scores in the anomaly score distribution served by the teacher, enabling GraphNC to learn more compact normal representations. GraphNC is comprehensively verified by empirical results on six GAD datasets, which show that GraphNC consistently enhances performance of three different types of teacher models, and it obtains better performance if the teacher model is stronger.

**Limitation and Future work.** A potential limitation of GraphNC is that it relies on an assumption that the pre-trained teacher model performs fairly well to provide informative supervision signals for the ScoreDA component. Future work will explore other sources, such as some labeled anomaly data or expert knowledge, as supplementary supervision to reduce its reliance on the teacher model.

## REPRODUCIBILITY STATEMENT

We have taken several steps to ensure the reproducibility of our work. The datasets used in our experiments are all publicly available, and the construction and analysis of the data are presented in detail in App. A. The implementation details, including model configurations and training hyperparameters, are thoroughly documented in Sec. 5. The source code has been submitted as supplementary materials.

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

## A  DETAILED DESCRIPTION OF DATASETS

The key statistics of the datasets are presented in Table 4. A detailed introduction of these datasets is given as follows.

- Amazon (Dou et al., 2020): It is a co-review network obtained from the Musical Instrument category on Amazon.com. There are also three relations: U-P-U (users reviewing at least one same product), U-S-U (users having at least one same star rating within one week), and U-V-U (users with top-5% mutual review similarities).

- T-Finance (Tang et al., 2022): It is a financial transaction network where the node represents an anonymous account and the edge represents two accounts that have transaction records. Some attributes of logging like registration days, logging activities, and interaction frequency, etc, are used as the features of each account. Users are labeled as anomalies if they fall into categories such as fraud, money laundering, or online gambling

- Reddit (Kumar et al., 2019): It is a user-subreddit graph which captures one month's worth of posts shared across various subreddits at Reddit. The node represents the users, and the text of each post is transformed into a feature vector and the features of the user and subreddits are the feature summation of the post they have posted. The anomalies are the used who have been banned by the platform

- YelpChi (Dou et al., 2020): Similar to Amazon-all, YelpChi-all incorporates three types of edges: R-U-R (reviews written by the same user), R-S-R (reviews of the same product with identical star ratings), and R-T-R (reviews of the same product posted within the same month). YelpChi-all is then constructed by merging these different relations into a single unified relation (Chen et al., 2022a; Qiao & Pang, 2023).

- Tolokers (McAuley et al., 2015): It is obtained from the Toloka crowdsourcing platform, where the node represents the person who has participated in the selected project, and the edge represents two workers work on the same task. The attributes of the node are the profile and task performance statistics of workers

- Photo (Shchur et al., 2018): It is an Amazon co-purchase network where nodes represent products and edges indicate co-purchase relationships. Each node is described by a bag-of-words representation derived from user reviews

## B  DESCRIPTION OF BASELINES

A more detailed introduction of the nine competing GAD models is given as follows.

- DOMINANT (Ding et al., 2019): It uses a three-layer GCN to reconstruct both structure and attributes, and the resulting reconstruction error on both structure and attribute is taken as the anomaly score.

- AnomalyDAE (Fan et al., 2020): It incorporates graph attention into the GNN-based structure and attribute encoders, where the anomaly score is derived from the reconstruction error.

- OCGNN (Wang et al., 2021): It applies a one-class approach on the GNN, aiming to combine the representational power of the GNN with the anomaly detection capability of one-class classification. The anomaly score is computed by measuring the distance of each point from the center.

- AEGIS (Ding et al., 2021): It leverages a generative adversarial network (GAN), where the generator is designed to produce pseudo anomalies, while the discriminator distinguishes between genuine normal nodes and the generated anomalies.

- GAAN (Chen et al., 2020): It leverages a generative adversarial network where fake graph nodes are generated. Then the covariance matrix for real nodes and fake nodes are computer to enhace the node. Finally, a discriminator is trained to recognize whether two connected nodes are from a real or fake node.

- TAM (Qiao & Pang, 2023): It first reveals the 'one class homophily' and introduces the new anomaly measure score 'local node affinity'. As the local node affinity can not be

Table 4: Key statistics of GAD datasets.

| Datasets | Amazon | T-Finance | Reddit | YelpChi | Tolokers | Photo |
|---|---|---|---|---|---|---|
| #Nodes | 11,944 | 39,357 | 10,984 | 45,941 | 11,758 | 7,484 |
| #Edges | 4,398,392 | 21,222,543 | 168,016 | 3,846,979 | 519,000 | 119,043 |
| #Attributes | 25 | 10 | 64 | 32 | 10 | 745 |
| Anomaly | 9.5% | 4.6% | 3.3% | 14.52% | 21.8% | 4.9% |

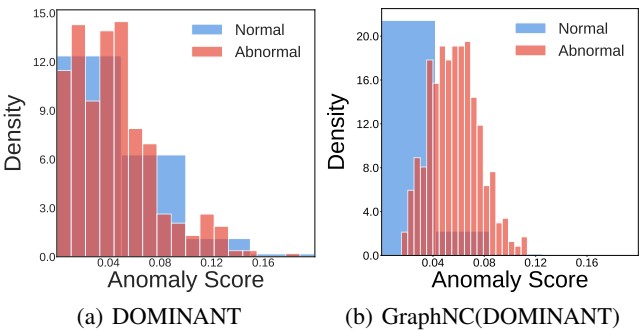

(a) DOMINANT  (b) GraphNC(DOMINANT)

Figure 5: The score distribution of DOMINANT along the corresponding NomrDR enabled DOMINANT on Amazon (Dou et al., 2020)

directly obtained based on the embedding, it is optimized on truncated graphs where non-homophily edges are removed iteratively. The local node affinity is calculated on the learned representations on the truncated graphs.

- GADNR (Roy et al., 2024): It employs a GNN-based autoencoder to reconstruct each node's attributes, degree, and Gaussian-approximated neighbor feature distribution, and uses the weighted sum of these reconstruction losses as the anomaly score.

- ADA-GAD (He et al., 2024): It adopts a two-stage anomaly-denoised autoencoder framework that first pretrains on anomaly-reduced augmented graphs and then retrains decoders on the original graph, taking the final reconstruction error with a distribution regularization term as the anomaly score.

- GGAD (Qiao et al., 2024): It employs two priors related to anomalies, asymmetric local affinity and egocentric closeness, to generate pseudo anomaly nodes that can well simulate the real abnormal nodes. Then, a binary one-class classifier is trained based on the existing normal nodes and generated outliers for semi-supervised GAD.

## C  ADDITIONAL EXPERIMENTAL RESULTS

### C.1  SORE DISTRIBUTION VISUALIZATION ON OTHER TEACHER MODEL

In this section, we visualized the score distributions of the reconstruction-based method DOMINANT and the one-class classification method OCGNN, along with their corresponding GraphNC-enabled models on the Amazon. As shown in the Fig. 5 and 6, we observe that the GraphNC-enabled model achieves better separation between normal and abnormal classes, further demonstrating that GraphNC can substantially and consistently enhance performance.

### C.2  ABLATION STUDY ON OTHER TEACHER MODELS

We also conduct the ablation study on the GraphNC-enabled other two teacher models to demonstrate the effectiveness of each module in the GraphNC. We employ the reconstruction-based method, DOMINANT, and one-class classification-based method, OCGNN as the teacher model and make a comparison with the variant of GraphNC. The experimental results are shown in the

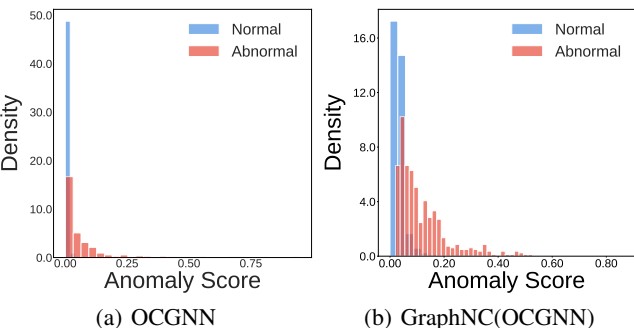

(a) OCGNN  (b) GraphNC(OCGNN)

Figure 6: The score distribution of OCGNN along the corresponding GraphNC enabled OCGNN on Amazon (Dou et al., 2020)

Table. 5 We observe that GraphNC outperforms the corresponding variant on most of the datasets for both DOMINANT and OCGNN. This further demonstrates the flexibility and effectiveness of GraphNC in enhancing existing semi-supervised methods.

Table 5: AUROC and AUPRC results comparison of the proposed method GraphNC and its variants.

| Metric | Method | Amazon | T-Finance | Reddit | YelpChi | Tolokers | Photo | Avg. |
|--------|--------|--------|-----------|--------|---------|----------|-------|------|
| | | DOMINANT | | | | | | |
| | OT+ScoreDA+NormReg | **0.8936** | **0.7985** | 0.5260 | **0.6543** | **0.6977** | **0.6829** | **0.7088** |
| | OT+ScoreDA | 0.7103 | 0.7069 | 0.4623 | 0.5125 | 0.5342 | 0.6384 | 0.5941 |
| | OT+NormReg-Finetune | 0.6054 | 0.4920 | 0.5192 | 0.6299 | 0.6258 | 0.5188 | 0.5651 |
| | OT+NormReg | 0.7320 | 0.5041 | **0.5968** | 0.6315 | 0.5943 | 0.5128 | 0.5952 |
| | OT | 0.8867 | 0.6167 | 0.5194 | 0.6314 | 0.5121 | 0.5314 | 0.6162 |
| AUROC | | OCGNN | | | | | | |
| | OT+ScoreDA+NormReg | **0.9255** | **0.6819** | **0.6093** | **0.6447** | 0.6069 | **0.7068** | **0.6956** |
| | OT+ScoreDA | 0.9158 | 0.4101 | 0.5235 | 0.6411 | **0.6134** | 0.6913 | 0.6325 |
| | OT+NormReg-Finetune | 0.8274 | 0.5960 | 0.4939 | 0.4738 | 0.5309 | 0.5588 | 0.5801 |
| | OT+NormReg | 0.9095 | 0.6113 | 0.5820 | 0.6248 | 0.5245 | 0.6478 | 0.6499 |
| | OT | 0.8810 | 0.5742 | 0.5622 | 0.6155 | 0.4803 | 0.6461 | 0.6265 |
| | | DOMINANT | | | | | | |
| | OT+ScoreDA+NormReg | **0.7290** | **0.3703** | 0.0414 | **0.2375** | **0.3465** | **0.1526** | **0.2795** |
| | OT+ScoreDA | 0.1466 | 0.1759 | 0.0312 | 0.1581 | 0.2445 | 0.1274 | 0.1472 |
| | OT+NormReg-Finetune | 0.0750 | 0.0415 | 0.0330 | 0.2292 | 0.2793 | 0.1122 | 0.1283 |
| | OT+NormReg | 0.1094 | 0.0424 | **0.0486** | 0.2307 | 0.2445 | 0.1066 | 0.1303 |
| | OT | 0.7289 | 0.0542 | 0.0414 | 0.2237 | 0.2217 | 0.1283 | 0.2330 |
| AUPRC | | OCGNN | | | | | | |
| | OT+ScoreDA+NormReg | **0.7372** | **0.0748** | **0.0512** | **0.2116** | 0.2814 | **0.1823** | **0.2564** |
| | OT+ScoreDA | 0.6298 | 0.0350 | 0.0327 | 0.2112 | **0.2915** | 0.1782 | 0.2297 |
| | OT+NormReg-Finetune | 0.4785 | 0.0554 | 0.0326 | 0.1382 | 0.2433 | 0.1093 | 0.1762 |
| | OT+NormReg | **0.7015** | 0.0567 | 0.0444 | 0.2036 | 0.2415 | 0.1525 | 0.2333 |
| | OT | 0.6895 | 0.0492 | 0.0400 | 0.1937 | 0.2138 | 0.1501 | 0.2227 |

## C.3 THE PERFORMANCE OF GRAPHNC WHEN THE TEACHER PERFORMS POORLY

To evaluate the performance of GraphNC when the teacher performs poorly (AUROC $\leq 0.5$), we inject anomalies into the labeled normal nodes to deliberately degrade the teacher's performance. When the teacher model performs poorly, we still apply score distribution alignment and NormReg to achieve normality calibration. The experimental results using GGAD as teacher model are shown in Table. 6.

Table 6: AUROC and AUPRC results of GraphNC and GGAD.

| Metric | Method | Dataset | | | | | | |
|--------|--------|---------|-----------|--------|--------|----------|--------|--------|
|        |        | Amazon  | T-Finance | Reddit | YelpChi | Tolokers | Photo  | Avg.   |
| AUROC  | GGAD    | 0.6336  | 0.5132    | 0.4483 | 0.5130 | 0.4504   | 0.4465 | 0.5008 |
|        | GraphNC | **0.7769** | **0.7450** | **0.5303** | **0.5685** | **0.6291** | **0.5970** | **0.6411** |
| AUPRC  | GGAD    | 0.1171  | 0.0443    | 0.0276 | 0.1603 | 0.1946   | 0.0856 | 0.1049 |
|        | GraphNC | **0.2274** | **0.0930** | **0.0375** | **0.1823** | **0.2930** | **0.1145** | **0.1580** |

Table 7: The comparison with GraphNC under the joint training with the teacher.

| Metric | Method | Dataset | | | | | | |
|--------|--------|---------|-----------|--------|--------|----------|--------|--------|
|        |        | Amazon  | T-Finance | Reddit | YelpChi | Tolokers | Photo  | Avg.   |
| AUROC  | GraphNC-Joint | 0.9391 | 0.7631 | 0.5273 | 0.5654 | 0.4626 | 0.7441 | 0.6669 |
|        | GraphNC | **0.9613** | **0.8340** | **0.6420** | **0.6630** | **0.6505** | **0.7693** | **0.7534** |

## C.4 THE COMPARISON WITH GRAPHNC UNDER THE JOINT TRAINING WITH THE TEACHER

We have explicitly articulated that GraphNC is built upon a pre-trained semi-supervised GAD model (as the teacher model), indicating a sequential training of the teacher model and the GraphNC model (the student model). There are three fundamental reasons for this GraphNC design. i) Since GraphNC involves a teacher-student learning component (ScoreDA), it is assumed that we have a pre-trained teacher model so that the teacher can provide useful supervision signals for the student. ii) Motivated by the high false positives and false negatives in the teacher models, GraphNC (the student model) is designed specifically to reduce these two types of errors—validated through both theoretical analysis and empirical results—while leveraging the knowledge learned in the teacher models. iii) The separate training can provide a stable optimization trajectory for optimizing the student model. In contrast, joint training often obscures the optimal direction due to the large shifting in the supervision signals from the teacher model. For completeness, we also provide the results of GraphNC under the joint-training setting for comparison, as shown in Table. 7, which demonstrate a largely degraded performance if the training of the teacher and the student models is done jointly.

## C.5 ANALYSIS ON TRUE ANOMALOUS INSTANCES RECONGINTION

To offer more insights into the dependency between our model and the teacher model, we provide statistics for the number of anomalous instances which are accurately detected by the teacher (GGAD) versus those detected by GraphNC in Table. 8. We can observe that the number of anomalous instances accurately detected by GraphNC is consistently and substantially higher than the teacher model across different datasets where the teacher model's performance varies largely. This indicates that GraphNC can go far beyond the supervision from the teacher model to detect anomalies that the teacher model missed. We attribute this capability to the NormReg component and its synergy with the ScoreDA component.

## C.6 STATISTICS OF THE AVERAGE AUROC IMPROVEMENT OF GRAPHNC W.R.T. DIFFERENT ACCURACY LEVELS

We summarize the average improvement of GraphNC over the teacher model across different teacher score ranges, as presented in Table. 9.

## D THEORETICAL ANALYSIS

**Theorem 1.** *Let $\sigma_{\mathcal{T}}^2$ and $\sigma_{\mathcal{S}}^2$ denote the variance of the anomaly scores yielded by the teacher model and the student model in the normal class, respectively, then under our GraphNC framework, the student model achieves shrinking score variance (i.e., $\sigma_{\mathcal{S}}^2 < \sigma_{\mathcal{T}}^2$) by minimizing $\mathcal{L}_{NormReg}$ while applying $\mathcal{L}_{ScoreDA}$, thereby reducing False Positive Rate (FPR) and False Negative Rate (FNR).*

Table 8: The number of true anomalous instances that are accurately detected by the teacher (GGAD) versus those detected by GraphNC.

| Method | Dataset | | | | | | Avg. |
|--------|---------|---------|--------|--------|----------|-------|------|
|        | Amazon  | T-Finance | Reddit | YelpChi | Tolokers | Photo |      |
| GGAD   | 423     | 707     | 55     | 1186   | 378      | 150   | 483  |
| GraphNC | **438** | **757** | **71** | **1352** | **455** | **214** | **548** |

Table 9: Statistics of the average AUROC improvement of GraphNC w.r.t. different accuracy levels of supervision from the teacher model GGAD.

| Teacher (GGAD) | 0.4-0.5 | 0.5-0.6 | 0.6-0.7 | 0.7-0.9 |
|----------------|---------|---------|---------|---------|
| GraphNC        | 0.1345  | 0.1128  | 0.0764  | 0.0204  |

*Proof.* We denote the teacher model's score on the normal class by the random variable $T$ (*i.e.*, $S_T|y=0) \sim (\mu_0, \sigma_T^2)$, which follows a sub-Gaussian distribution, with a mean of $\mu_0$ and a variance of $Var(T) = \sigma_T^2$. The law of total variance is given by

$$Var(T) = Var(\mathbb{E}[T|\mathbf{X}]) + \mathbb{E}(Var(T|\mathbf{X})), \tag{6}$$

where $\mathbf{X}$ is the input feature. The optimal solution of the student model $\mathcal{F}_S$ learned through soft score distillation with the input feature is $\mathcal{F}_S^*(\mathbf{X}) = \mathbb{E}[T|\mathbf{X}]$, then we have

$$Var(\mathcal{F}_S^*(\mathbf{X})) = Var(\mathbb{E}[T|\mathbf{X}]) \leq Var(T) = \sigma_T^2. \tag{7}$$

By optimizing the $\mathcal{L}_{ScoreDA}$, the variance of the predicted scores of the student model on normal nodes is usually not larger than that of the teacher model.

Furthermore, the designed $\mathcal{L}_{NormReg}$ loss further shrinks the intra-class variance of normal classes on the student model. $\mu_0$ can be regarded as the potential true center of the normal class, and then the representation of each normal node $v_i$ in the two views of the student model can be written as $h_i = \mu_0 + \epsilon_i$, $\widetilde{h}_i = \mu_0 + \widetilde{\epsilon}_i$, where $\epsilon_i$ and $\widetilde{\epsilon}_i$ are the perturbations or noise introduced by two different views. We assume that the perturbations in both views are statistically independent and have zero mean, *i.e.*, $\mathbb{E}[\epsilon_i] = \mathbb{E}[\widetilde{\epsilon}_i] = 0$. By substituting $h_i$ and $\widetilde{h}_i$ into the $\mathcal{L}_{NormReg}$, we can redefine it as

$$\mathcal{L}_{NormReg} = \frac{1}{|\mathcal{V}_l|} \sum_{v_i \in \mathcal{V}_l} \|\epsilon_i - \widetilde{\epsilon}_i\|_2^2 \tag{8}$$

By unfolding the loss and then computing its expectations, we obtain

$$\mathbb{E}[\|\epsilon_i - \widetilde{\epsilon}_i\|_2^2] = \mathbb{E}[\|\epsilon_i\|_2^2 + \|\widetilde{\epsilon}_i\|_2^2 - 2\langle\epsilon_i, \widetilde{\epsilon}_i\rangle]$$
$$= \mathbb{E}[\|\epsilon_i\|_2^2] + \mathbb{E}[\|\widetilde{\epsilon}_i\|_2^2] - 2\langle\mathbb{E}[\epsilon_i], \mathbb{E}[\widetilde{\epsilon}_i]\rangle \tag{9}$$

Since $\epsilon_i$ and $\widetilde{\epsilon}_i$ are independent, and $\langle\mathbb{E}[\epsilon_i], \mathbb{E}[\widetilde{\epsilon}_i]\rangle = 0$. We have,

$$\mathbb{E}[\|\epsilon_i - \widetilde{\epsilon}_i\|_2^2] = \mathbb{E}[\|\epsilon_i\|_2^2] + \mathbb{E}[\|\widetilde{\epsilon}_i\|_2^2] = \sigma_\epsilon^2 + \sigma_{\widetilde{\epsilon}}^2, \tag{10}$$

where $\sigma_\epsilon^2$ and $\sigma_{\widetilde{\epsilon}}^2$ are the population variances of the two views in the student model, respectively. By minimizing $\mathcal{L}_{NormReg}$, the node embeddings tend to be consistent across the two views (*i.e.*, the consistent variances pattern) and the population variance decreases, making the normal node embeddings more compactly clustered around $\mu_0$. It is known that $Var(\mathcal{F}_S^*(\mathbf{X}))$ corresponds to the variance of the normal class in the student model when considering only the distillation loss $\mathcal{L}_{ScoreDA}$. Therefore, when $\mathcal{L}_{NormReg}$ is taken into account, the variance of the normal class in the student model satisfies $\sigma_S^2 = Var(\mathcal{F}_S^*(\mathcal{H})) < Var(\mathcal{F}_S^*(\mathbf{X})) \leq \sigma_T^2$, where $\mathcal{H}$ is the output of the student model.

Since $S_T|y=0$ follows a sub-Gaussian distribution, there exists $\sigma_T$ such that for $t > 0$ it satisfies

$$\mathbb{P}(S_T - \mu_0 \geq t) \leq exp(-\frac{t^2}{2\sigma_T^2}) \tag{11}$$

Given an arbitrary fixed threshold $\tau$ ($\mu_0 < \tau$), if the predicted score of a sample exceeds this threshold, the sample is classified as abnormal. According to Eq. (11) and taking $t = \tau - \mu_0$, we have,

$$\mathbb{P}(S_s \geq \tau|y=0) \leq exp(-\frac{(\tau - \mu_0)^2}{2\sigma_S^2}), \quad \mathbb{P}(S_T \geq \tau|y=0) \leq exp(-\frac{(\tau - \mu_0)^2}{2\sigma_T^2}) \tag{12}$$

Table 10: Runtimes (in seconds) of training on the six datasets of GGAD and GraphNC.

| Method | Dataset | | | | | |
|--------|---------|-----------|--------|--------|----------|-------|
|        | Amazon  | T-Finance | Reddit | YelpChi | Tolokers | Photo |
| GGAD   | 658     | 9345      | 368    | 480     | 850      | 106   |
| GraphNC | 60     | 3050      | 900    | 4200    | 19       | 89    |

Table 11: Runtimes (in seconds) of inference on the six datasets of four methods.

| Method | Dataset | | | | | |
|--------|---------|-----------|--------|--------|----------|--------|
|        | Amazon  | T-Finance | Reddit | YelpChi | Tolokers | Photo |
| DOMINANT | 37.8001 | 69.3396 | 1.3239 | 34.1469 | 2.7525 | 1.2359 |
| OCGNN | 0.1141 | 1.1454 | 0.0944 | 1.4889 | 0.1067 | 0.0502 |
| GGAD | 0.0991 | 1.0896 | 0.0867 | 1.4931 | 0.1023 | 0.0508 |
| GraphNC | 0.0951 | 1.0929 | 0.0785 | 1.5323 | 0.0934 | 0.0403 |

Since $\sigma_S^2 < \sigma_T^2$, $\mathbb{P}(S_s \geq \tau | y = 0) < \mathbb{P}(S_T \geq \tau | y = 0)$. It indicates that the probability of the student model wrongly identifying an abnormal node as a normal node is lower than that of the teacher model, thereby reducing FNR.

Similarly, if the predicted score is below this threshold, the node is classified as normal. We compute the probabilities of the student model ($\mathbb{P}(S_s < \tau | y = 0)$) and the teacher model ($\mathbb{P}(S_T < \tau | y = 0)$) predicting it as a normal node as follows,

$$\mathbb{P}(S_s < \tau | y = 0) = 1 - \mathbb{P}(S_s \geq \tau | y = 0), \quad \mathbb{P}(S_T < \tau | y = 0) = 1 - \mathbb{P}(S_T \geq \tau | y = 0). \quad (13)$$

Since $\mathbb{P}(S_s \geq \tau | y = 0) < \mathbb{P}(S_T \geq \tau | y = 0)$, we have,

$$1 - \mathbb{P}(S_s \geq \tau | y = 0) \geq 1 - \mathbb{P}(S_T \geq \tau | y = 0). \quad (14)$$

Finally, we derive $\mathbb{P}(S_s < \tau | y = 0) > \mathbb{P}(S_T < \tau | y = 0)$, which can equivalently be expressed as $\mathbb{P}(S_s < \tau | y = 1) < \mathbb{P}(S_T < \tau | y = 1)$, given that the anomaly detection task uses binary labels ($y = 0$ for normal and $y = 1$ for abnormal). It indicates that the probability of misclassifying a normal node as abnormal is lower in the student model than in the teacher model, leading to a reduced FPR. $\qquad\square$

## E  COMPUTATIONAL EFFICIENCY ANALYSIS

### E.1  TIME COMPLEXITY ANALYSIS

In this section, we analyze the time complexity of GraphNC. Since it is designed as a plug-in framework that can be integrated with various teacher models to enhance GAD, we focus here only on the complexity of the additional module itself. We employ the combination of GCN and one MLP layer as the student model in GraphNC. The GCN takes $\mathcal{O}(EM + NMd)$, where $E$ is the number of edges in the graph, $N$ is the number of nodes, $M$ is the dimension of the attribute, and $d$ is the dimension of representation. The MLP used for feature transformation in GraphNC is $\mathcal{O}(NMd)$. In GraphNC, ScoreDA aims to align the student score with the output score distribution, which will take $O(N)$. The NormReg aiming to minimize the distance between two views, which will also take $O(N)$ The total complexity is $\mathcal{O}(EM + 2NMd + 2N) + \mathbf{T}$, where $\mathbf{T}$ is the time complexity of the corresponding teacher model.

### E.2  RUNTIME RESULTS

We provide a runtime comparison between the standalone teacher model and the model augmented with GraphNC below. The training and inference time comparison are shown in Table. 10 and 11. It is clear that GraphNC is highly efficient for training, as its training process builds directly upon the pretrained teacher model. In addition, GGAD and GraphNC are both very efficient for inference.

## F  ALGORITHM

The algorithm of GraphNC is summarized in Algorithm 1

**Algorithm 1** GraphNC

**Input**: Graph $\mathcal{G} = (\mathcal{V}, \mathcal{E}, \mathbf{X})$; Pre-trained Semi-supervised GAD Model $\mathcal{F}_\mathcal{T}$; Student Model $\mathcal{F}_S$ with parameters $\Phi$; Labeled Normal nodes $\mathcal{V}_l \subset \mathcal{V}$; Number of training epochs $E$; Trade-off weight $\alpha$.

**Output**: Anomaly score $y_i^\mathcal{S}$ for each node $v_i \in \mathcal{V}$.

1: Initialize parameters $\Theta$ for the student model $\mathcal{F}_S$.
2: Obtain the anomaly scores distribution $\mathcal{Y}^\mathcal{T} = \mathcal{F}_\mathcal{T}(\mathbf{X}, \mathcal{G}; \Theta) = \{y_1^\mathcal{T}, y_2^\mathcal{T}, ..., y_N^\mathcal{T}\}$ from the teacher model.
3: Create an augmented feature matrix $\mathbf{X}'$ by applying RandomMask to features of nodes in $\mathcal{V}_l$.
4: **for** $e = 1$ to $E$ **do**
5:    **for** $v_i$ in $\mathcal{V}$ **do**
6:       Obtain the representation for node $v_i$: $\mathbf{h}_i^\mathcal{S} \leftarrow \mathcal{F}_{GNN}(\mathbf{x}_i, \mathcal{G}, \mathbf{X}; \Omega)$.
7:       Obtain the augmented representation for node $v_i$: $\widetilde{\mathbf{h}}_i^\mathcal{S} \leftarrow \mathcal{F}_{GNN}(\tilde{\mathbf{x}}_i, \tilde{\mathcal{G}}, \tilde{\mathbf{X}}; \Omega)$.
8:       Obtain the anomaly score of student model for node $v_i$: $y_i^\mathcal{S} = \mathcal{F}_{MLP}(\mathbf{h}_i^\mathcal{S}; \phi)$.
9:    **end for**
10:    *// — Anomaly Score Distribution Alignment (ScoreDA) —*
     $\mathcal{L}_{ScoreDA} = \frac{1}{|\mathcal{V}|} \sum_{v_i \in \mathcal{V}} \|y_i^\mathcal{S} - y_i^\mathcal{T}\|_2^2$.
11:    *// — Perturbation-based Regularization (NormaReg) —*
12:    $\mathcal{L}_{NormReg} = -\frac{1}{|\mathcal{V}_l|} \sum_{v_i \in \mathcal{V}_l} \left\| \mathbf{h}_i^\mathcal{S} - \widetilde{\mathbf{h}}_i^\mathcal{S} \right\|_2^2$.
13:    *// — Total Loss Function —*
14:    $\mathcal{L}_{GraphNC} \leftarrow \mathcal{L}_{ScoreDA} + \alpha \cdot \mathcal{L}_{NormReg}$.
15:    Minizing $\mathcal{L}_{GraphNC}$ and update parameters of student model, $\Phi = \{\Omega, \phi\}$ by gradient descending .
16: **end for**
17: **return** Anomaly scores $S(v_i) = \mathcal{F}_\mathcal{S}(v_i, \mathbf{x}_i, \mathcal{G}, \Phi^*)$, for each node $v_i \in \mathcal{V}$

