# OpenReview forum: "Normality Calibration in Semi-supervised Graph Anomaly Detection"
_ICLR.cc/2026/Conference — Submitted to ICLR 2026_

### Official Review · Reviewer_GNkY · 2025-10-29

**Soundness:** 3
**Presentation:** 3
**Contribution:** 2
**Rating:** 2
**Confidence:** 5

**Summary:**

The paper proposes GraphNC, a teacher–student framework for semi-supervised graph anomaly detection (GAD). It introduces two main components: Score Distribution Alignment (ScoreDA) aligns the student’s anomaly scores with the teacher’s distribution, aiming to calibrate the notion of “normality.”  Perturbation-guided Normality Regularization (NormReg) applies consistency regularization to labeled normal nodes, enhancing intra-class compactness.

**Strengths:**

* The method can be applied on top of various existing GAD models, showing flexibility.

**Weaknesses:**

1. The proposed method is essentially a straightforward combination of existing ideas. The teacher model is drawn directly from the baselines, while the student model is a simple contrastive learning framework augmented by an extra loss term (L_{ScoreDA}). There is little genuine innovation in either architecture or learning strategy. Yet, the method achieves surprisingly high performance, which raises questions about where the actual gain comes from — this will be discussed further below.

2. The authors train the teacher model on each dataset exactly as in the original papers, under the same semi-supervised setting. Then, on the same dataset, they train the student model by aligning its predictions with the teacher’s anomaly scores. From the loss formulation, it is clear that the teacher’s predicted scores are treated as ground truth, without considering possible teacher errors or noise. Consequently, the overall performance depends entirely on how good the teacher is.
Table 5 confirms this dependency: when the teacher performs poorly on a dataset, the proposed method can even hurt performance instead of improving it.

3. Conceptual inconsistency with the teacher–student paradigm and semi-supervised learning:
This experimental design violates the fundamental purpose of knowledge distillation. In typical distillation, the goal is for a smaller or cheaper student model to approximate or even surpass a strong teacher using the same data or fewer resources. However, here the student is trained after a full teacher pretraining phase on the same dataset, effectively making the process redundant and inefficient.
Moreover, when the teacher model already achieves very high accuracy (e.g., We regard teacher scores as near-supervised labels when (i) AUPRC ≥ 0.8 on datasets with anomaly rate ≤ 1%, (ii) Precision@Top-1% ≥ 90%, and (iii) OOF-estimated label noise within the top-scored set ≤ 5%), this pipeline can no longer be considered semi-supervised. The student is simply learning from near-perfect labels, making the task closer to a fully supervised binary classification problem rather than genuine graph anomaly detection. This undermines the originality and validity of the proposed framework.

**Questions:**

* Did you use an Out-of-Fold (OOF) protocol to avoid leakage from teacher to student?

* Equation (3) sign/direction appears inconsistent with the text (minimize vs. maximize), and Algorithm 1 repeats the issue.  Equation (3) defines L_NormReg =  -XXX. , while the text says it minimizes the discrepancy between representations. This negative sign contradicts both the description and Algorithm 1 (which minimizes the total loss). Could the authors clarify whether L_NormReg is minimized or maximized in implementation, and correct the sign accordingly?

---

> ### Author Response · Authors · 2025-11-20
>
> We sincerely appreciate your constructive and positive comments on our methodology design. Please see our response to your comments one by one below.
>
> >**Weakness #1:**  The innovation in either architecture or learning strategy
>
> Thank you very much for the comments. Please refer to the response to **Reviewer XpiH Weakness #1**, where we illustrate the significant technical and theoretical contributions of GraphNC from three fronts. We would be grateful if you could kindly take these three contributions into account when assessing the novelty of this work.
>
>
> >**Weakness #2:** Semi-supervised setting and the teacher dependency when the teacher performs poorly
>
> Thank you very much for your comments. We would like to clarify that i) the ground-truth labels used in our setting are a subset of **labeled normal samples** prior to training, and ii) the proposed model's effectiveness is contributed by not only the prediction scores from the teacher model but also the labeled normal node set. In particular, the teacher’s predictions are used as pseudo-labels for score distribution alignment (the ScoreDA component) only; we also have another component, NormReg, both of which work complementarily to each other, ensuring that GraphNC can i) achieve stable improvement in various semi-supervised learning scenarios, and ii) gain better performance when the teacher's predictions are more accurate.  For example, our model does not collapse when the teacher model fails to work. For strong empirical justification of this point, please refer to our response to **Reviewer tinT Weakness #1**, where we evaluate GraphNC’s performance when the teacher performs poorly and provide an analysis showing why GraphNC can still deliver notable improvements even under highly inaccurate teacher supervision. In addition, from Tables 2 and 3, the experimental results also show that GraphNC consistently improves—rather than degrades—the performance of the teacher models. Furthermore, the component analysis in Table 5 demonstrates that the combination of ScoreDA and NormReg effectively enhances performance, even in cases where the teacher performs poorly on certain datasets.

---

> ### Author Response · Authors · 2025-11-20
>
> >**Weakness #3 and Question #1:** Conceptual inconsistency with the teacher–student paradigm and semi-supervised learning, this pipeline can no longer be considered semi-supervised and Out-of-Fold (OOF) protocol for avoiding leakage
>
> Thank you very much for the comments.  We would like to clarify that the teacher model is trained exclusively on a small set of labeled normal samples (15\% of the full normal node set in a graph), whereas the student model is trained on the entire dataset, including both the labeled normal samples and the unlabeled data. Learning from large unlabeled data and a small set of labeled data is a well-established semi-supervised problem, rather than a fully-supervised problem. The teacher’s predictions are used solely as **pseudo-labels** (not human-annotated labels) in our score distribution alignment, and these pseudo labels are not confirmed by humans in any way, and thus, this does not change the semi-supervised nature of  the studied problem.
> In addition, we also vary the random seeds in sampling the labeled nodes and repeat all experiments 5 times independently, and the main results reported in the paper are the average performance across five runs. In the inference, the student model is evaluated on the unlabeled node set. Thus, our training and evaluation protocols are strictly adhered to a semi-supervised setting.
>
> Moreover, from the experiments, we observe that not all methods perform consistently well across all datasets, especially the reconstruction-based and one-class classification methods, which tend to show weaker or unstable performance under this setting. This also indicates that GraphNC does not rely on a well-performing teacher model. Following the suggestion of **Reviewer tinT Weakness #1** ,  where we evaluate GraphNC’s performance under poorly performing teacher models and provide an analysis explaining why GraphNC can still achieve corresponding improvements even when the teacher provides highly inaccurate supervision. We specifically designed an experiment where the teacher model’s AUROC is intentionally degraded to below 0.5. The results clearly show that GraphNC can still yield performance improvements even under such poor teacher supervision.  Besides, to offer more insights into the dependency between our model and the teacher model, we provide statistics for the number of anomalous instances which are accurately detected by the teacher (GGAD) versus those detected by GraphNC in Table A1. We can observe that the number of anomalous instances accurately detected by GraphNC is consistently and substantially higher than the teacher model across different datasets, where the teacher model's performance varies largely. This indicates that GraphNC can go far beyond the supervision from the teacher model to detect anomalies that the teacher model missed. We attribute this capability to the NormReg component and its synergy with the ScoreDA component.
>
> ```
> Table A1. The number of true anomalous instances that are accurately detected by the teacher (GGAD) versus those detected by GraphNC.
> ```
> | Methods       | Amazon | T-Finance | Reddit | YelpChi | Tolokers | Photo |
> |---------------|--------|-----------|--------|---------|----------|-------|
> | Teacher (GGAD) | 423    | 707       | 55     | 1186    | 378      | 150   |
> | GraphNC       | **438**    | **757**       |**71**     | **1352**    | **455**      | **214**   |
>
>
> >**Question #2:** Equation (3) sign/direction appears inconsistent.
>
> Thank you very much for pointing this out. The negative sign in the equation is indeed a typographical error in the manuscript. We confirm that the implementation in our released code uses the correct formulation, and all reported results are based on the correct version. NormReg is designed to regularize the graph normality in the representation space, making the representations of normal nodes more compact by minimizing a perturbation-guided consistency loss solely on the labeled nodes. Our code correctly implements this minimization objective, and therefore, the typo in the manuscript does not affect any experimental results or conclusions.

---

> > ### Comment · Reviewer_GNkY · 2025-11-21
> >
> > Thank the author for their thoughtful response. However, it has not convinced me. The author avoided two questions: why the teacher model and student model were trained separately on the same dataset, and the motivation for distillation learning in this paper (save time or parameters).
> >
> > Therefore, I maintain my original score.

---

> > > ### Author Response · Authors · 2025-11-23
> > >
> > > We're pleased that you found our response thoughtful. We really appreciate your further comments. Please find our response to your follow-up questions as follows.
> > >
> > > >**Follow-up question with Question #1:**  Why were the teacher model and student model trained separately on the same dataset
> > >
> > > **Training on the same dataset.** First of all, we would like to clarify that most node-level graph machine learning tasks, including the task we explore in this work, is a transductive learning problem, i.e., training (and pre-training) and testing on the same dataset. Following this paradigm, both the student model (GraphNC) and the teacher model (the pre-trained model) are trained on the same dataset.
> > >
> > > **Why were they trained separately?** Secondly, we have explicitly articulated throughout the paper that GraphNC is built upon a pre-trained semi-supervised GAD model (as the teacher model), indicating a sequential training of the teacher model and the GraphNC model (the student model).  There are three fundamental reasons for this GraphNC design. i) Since GraphNC involves a teacher-student learning component (ScoreDA), it is assumed that we have a pre-trained teacher model so that the teacher can provide useful supervision signals for the student. ii) Motivated by the high false positives and false negatives in the teacher models, GraphNC (the student model) is designed specifically to reduce these two types of errors—validated through both theoretical analysis and empirical results—while leveraging the knowledge learned in the teacher models. iii) The separate training can provide a stable optimization trajectory for optimizing the student model. In contrast, joint training often obscures the optimal direction due to the large shifting in the supervision signals from the teacher model. For completeness, we also provide the results of GraphNC under the joint-training setting for comparison, as shown in Table A2, which demonstrate a largely degraded performance if the training of the teacher and the student models is done jointly.
> > >
> > > ```
> > > Table A2. The comparison with GraphNC under the joint training with the teacher.
> > > ```
> > > |    Methods    | Amazon | T-Finance | Reddit | YelpChi | Tolokers | Photo  |
> > > |---------------|--------|-----------|--------|---------|----------|--------|
> > > | GraphNC-Joint | 0.9391 |   0.7631  | 0.5273 | 0.5654  |  0.4626  | 0.7441 |
> > > |    GraphNC    | **0.9613** |   **0.8340**  | **0.6420** | **0.6630**  |  **0.6505**  | **0.7693** |

---

> > > > ### Author Response · Authors · 2025-11-23
> > > >
> > > > >**Follow-up question with Question #2:** The motivation for distillation learning in this paper (save time or parameters)
> > > >
> > > > Thank you for this follow-up question. The training time and parameters are not the key motivation behind our design of using a distillation learning framework; instead, our framework is designed to unleash the potential of both unlabeled and labeled node data in semi-supervised GAD.  Since existing semi-supervised GAD methods have shown promising results in leveraging the labeled node data, our framework GraphNC aims to develop novel contributions that can tap into these existing methods (via ScoreDA through a distillation learning structure) while at the same time mitigating some of their critical issues (i.e., high false positives and false negatives) via NormReg. The distillation learning paradigm offers a very effective framework for this purpose. **Thus, the distillation learning framework is not used to train a smaller or cheaper student model, as expected by the reviewer, but to better leverage the semi-supervised GAD data**. Please refer to *Lines 040-084* for detailed clarification of the motivations of these two components. We provide extensive empirical and theoretical results for the superiority of the proposed distillation learning framework and its remarkable plug-and-play ability in enabling various existing semi-supervised GAD methods.
> > > >
> > > > The training parameters and time cost are not a major concern here.  Regarding the model parameters, since GAD methods typically adopt a very small number of layers GNN architecture, the number of parameters remains comparable and relatively small for all the methods.
> > > >
> > > > In terms of training time cost, we acknowledge that the pre-training in the teacher model can indeed take some time, as shown in Table A3 in **Reviewer yumf Weakness #4**, particularly on some large datasets like T-Finance. However, the training can typically be done offline, and thus, our primary motivation is mainly on the detection accuracy. In addition, as shown in Table A3 below, GraphNC can even offer faster inference than the teacher models—particularly for reconstruction-based methods—which is often more crucial for real-world applications.
> > > >
> > > > ```
> > > > Table A3.  Runtimes (in seconds) of inference on the six datasets of four methods.
> > > > ```
> > > > |  Data   |  Amazon  | T-Finance | Reddit | YelpChi | Tolokers | Photo  |
> > > > |---------|----------|-----------|--------|---------|----------|--------|
> > > > | DOMINANT|  37.8001 |  69.3396  | 1.3239 | 34.1469 |  2.7525  | 1.2359 |
> > > > |  OCGNN  |  0.1141  |  1.1454   | 0.0944 | 1.4889  |  0.1067  | 0.0502 |
> > > > |  GGAD   |  0.0991  |  1.0896   | 0.0867 | 1.4931  |  0.1023  | 0.0508 |
> > > > | GraphNC |  0.0951  |  1.0929   | 0.0785 | 1.5323  |  0.0934  | 0.0403 |
> > > >
> > > >
> > > > In summary, as a plug-and-play module, GraphNC consistently improves the teacher models' performance even when the teacher performs poorly. It effectively reduces both the FPR and FNR of state-of-the-art methods (including teacher models) while having remarkable inference efficiency, highlighting the superiority and practical value of the proposed GAD-tailored distillation learning framework.
> > > >
> > > > We hope that the above reply helps address your follow-up questions. We will clarify these points in our final version. We're more than happy to engage in more discussion with you to address any further questions you may have. Thank you very much for helping improve our paper again!

---

> > > > > ### Author Response · Authors · 2025-11-28
> > > > >
> > > > > Dear Reviewer GNkY,
> > > > >
> > > > > We hope this message finds you well. We would like to express our sincere appreciation for the time you have spent reviewing our work and for your insightful comments. Your feedback has been invaluable in helping us strengthen and refine our manuscript.
> > > > >
> > > > > We would like to highlight the significant technical and theoretical contributions of GraphNC to GAD in the following three fronts.  i) The proposed distillation and consistency regularization methods are designed in a way to tailor specifically for the semi-supervised GAD problem; to the best of our knowledge, no prior work has been reported in this line. ii) Current semi-supervised GAD methods are focused on leveraging the labeled normal nodes, while GraphNC provides a novel framework orthogonal to enable those state-of-the-art to exploit the additional large-scale unlabeled data, making it the first framework that can plug in and play to enhance a diverse range of existing state-of-the-art detectors. iii) We show theoretically that the innovative way we adapt the distillation and consistency regularization techniques to GAD can effectively reduce False Positive Rate (FPR) and False Negative Rate (FNR) in existing state-of-the-art teacher models. We would be grateful if you could kindly take these three contributions into account when assessing the novelty of this work.
> > > > >
> > > > > We would be grateful if you could kindly take these three contributions into account when assessing the overall score of this work.
> > > > >
> > > > > As the discussion phase is nearing its end, if any aspects of our responses remain unclear or insufficient, we would be deeply grateful if you could kindly point them out. We will do our utmost to further improve and supplement the manuscript within the remaining time. Once again, thank you very much for your time, effort, and constructive evaluation of our submission. Your guidance is greatly appreciated.
> > > > >
> > > > > Sincerely,
> > > > > The Authors of Submission19076

---

### Official Review · Reviewer_XpiH · 2025-10-31

**Soundness:** 2
**Presentation:** 3
**Contribution:** 1
**Rating:** 4
**Confidence:** 4

**Summary:**

This paper presents GraphNC, a framework for semi-supervised graph anomaly detection that improves a student model by leveraging a pre-trained teacher model. Specifically, ScoreDA aligns the anomaly score distribution between teacher and student to enhance the capability of the student model, while NormReg, a perturbation-based masking regularization, mitigates the negative effect of inaccurate teacher outputs by enforcing compact representations of normal nodes.

**Strengths:**

1. The paper is clearly written, with a clear problem statement and a logically organized presentation.
  2. The experimental evaluation is fairly comprehensive, covering multiple datasets and various baseline methods.

**Weaknesses:**

1. The novelty of the paper is limited. the overall framework essentially follows a distillation paradigm, with the main contribution being the incorporation of a graph augmentation–based consistency regularization technique to improve the anomaly detection capability of the student model. However, the methodological innovation of the framework is relatively low, as it largely combines and applies existing ideas from distillation and consistency regularization.
2. The method assumes the teacher produces reasonably accurate scores for many nodes. If the teacher is poor, ScoreDA could harm performance.

**Questions:**

1. In Eq. (3), the loss term is defined with a negative sign, while the hyperparameter α is positive. This setup would effectively increase the discrepancy between original and augmented representations, which contradicts the explanation in lines 260–261 stating that NormReg minimizes this discrepancy. Could you clarify whether this is a typographical or conceptual mistake in the paper, or if I have misunderstood the formulation?
2. The primary objective of this work is to enable the student model to surpass the teacher model through a distillation-like approach. However, if the teacher model performs poorly on certain datasets, can the proposed framework still function effectively? It is recommended that the authors provide robustness experiments under such conditions or explicitly clarify this limitation in the paper.
3. In the NormReg module, random masking is applied to labeled normal nodes to generate augmented samples, and the model minimizes their representation discrepancy. However, masking may alter node features and deviate them from the normal distribution. How do the authors ensure that these augmented nodes remain normal? If some augmented samples lose their normality, wouldn’t minimizing their discrepancy reduce the model’s ability to distinguish anomalies?

---

> ### Author Response · Authors · 2025-11-20
>
> We sincerely appreciate your constructive and positive comments on our paper writing, presentation, and experiment analysis. Please see our response to your comments one by one below.
>
> >**Weakness #1:** The novelty of GraphNC
>
> While we acknowledge that  GraphNC takes inspiration from distillation and consistency regularization techniques, it makes significant technical and theoretical contributions to GAD in the following three fronts. i) The proposed distillation and consistency regularization methods are designed in a way to tailor specifically for the semi-supervised GAD problem; to the best of our knowledge, no prior work has been reported in this line. ii) Current semi-supervised GAD methods are focused on leveraging the labeled normal nodes, while GraphNC provides a novel framework orthogonal to enable those state-of-the-art to exploit the additional large-scale unlabeled data, making it the first framework that can plug in and play to enhance a diverse range of existing state-of-the-art detectors. iii) We show theoretically that the innovative way we adapt the distillation and consistency regularization techniques to GAD can effectively reduce False Positive Rate (FPR) and False Negative Rate (FNR) in existing state-of-the-art teacher models. We would be grateful if you could kindly take these three contributions into account when assessing the novelty of this work.
>
> >**Weakness #2 and Question #2:** If the teacher model performs poorly on certain datasets, can the proposed framework still function effectively?
>
> Thank you very much for the comments. Please refer to the response to **Reviewer tinT Weakness #1:**, where we evaluate GraphNC’s performance under poorly performing teacher models and provide an analysis explaining why GraphNC can still achieve corresponding improvements even when the teacher provides highly inaccurate supervision.
>
>
> >**Question #1:** Clarification on the sign of NormReg
>
> Thank you very much for pointing this out. The negative sign in the equation is indeed a typographical error in the manuscript. We confirm that the implementation in our released code uses the correct formulation, and all reported results are based on the correct version. NormReg is designed to regularize the graph normality in the representation space, making the representations of normal nodes more compact by minimizing a perturbation-guided consistency loss solely on the labeled nodes. Our code correctly implements this minimization objective, and therefore, the typo in the manuscript does not affect any experimental results or conclusions.
>
>
> >**Question #3:** How do the authors ensure that these augmented nodes remain normal? If some augmented samples lose their normality, wouldn’t minimizing their discrepancy reduce the model’s ability to distinguish anomalies?
>
> Thank you very much for the comments. We agree that a large masking ratio may indeed distort the underlying normality. As shown in Figures 4(c) and (d) of the main paper, increasing the mask ratio leads to a slight performance drop. For this reason, we adopt a relatively small mask ratio of 30\% as a fixed hyperparameter across all datasets. It is also important to note that the small random masks applied to normal nodes introduce only mild, localized perturbations—similar to random erasing or dropout in computer vision. Just as removing a few pixels from an image does not change its semantic identity, lightly masking a normal node does not alter its intrinsic normality. This form of normal-sample augmentation has been widely adopted across other data modalities as well, including tabular data [Ref1], time series [Ref2], and images [Ref3].
>
> Therefore, even if a few augmented views with small perturbations slightly deviate from their originals, they will not push normal nodes toward the anomalous region. Instead, the regularization improves the diversity of normal patterns and reinforces the compactness of the normal cluster while keeping anomalous nodes unaffected. Moreover, we theoretically show that such mild perturbations—combined with normality consistency maximization during score alignment—can effectively shrink the variance of the normal region and improve the separation between normal and anomalous nodes, thereby reducing both the false-positive and false-negative rates. On the other hand,  the empirical evidence also confirms that the combination of ScoreDA + NormReg consistently improves the AUROC and AUPRC across the benchmark datasets.
>
> - [Ref1] Tao, Shuting, et al. "Semanticmask: a contrastive view design for anomaly detection in tabular data." IJCAI, 2024.
> - [Ref2] Cho, Hyunsoo, Jinseok Seol, and Sang-goo Lee. "Masked Contrastive Learning for Anomaly Detection." IJCAI, 2021.
> - [Ref3] Fang, Yuchen, et al. "Temporal-frequency masked autoencoders for time series anomaly detection." ICDE, 2024.

---

> > ### Comment · Reviewer_XpiH · 2025-11-27
> >
> > Thank you for your detailed and thoughtful responses to my comments. While I find the responses helpful and they address some of my technical concerns, they do not substantially change my overall evaluation regarding the level of novelty and the dependence on teacher quality. Therefore, I will maintain my original score.

---

> > > ### Author Response · Authors · 2025-11-28
> > >
> > > Dear Reviewer XpiH
> > >
> > > Thank you for your response. We would like to clarify and respectfully argue that the current contribution score and overall scores may not fully reflect the significance of our work.
> > >
> > > GraphNC provides several contributions that, to our knowledge, have not been explored in prior GAD research:
> > >
> > > First,  while current semi-supervised GAD methods primarily rely on a limited set of labeled normal nodes [Ref1, Ref2, Ref3], GraphNC introduces a novel, orthogonal framework that enables existing state-of-the-art detectors to effectively leverage large-scale unlabeled data. This makes GraphNC the first plug-and-play enhancement module applicable across a wide range of GAD architectures.
> > >
> > > Second, to the best of our knowledge, no existing GAD method is designed explicitly to improve pre-trained GAD detectors, especially under the severe challenge that current semi-supervised GAD models often suffer from high FPR and FNR [Ref1, Ref2, Ref3] as illustrated in Figure 1 in the main paper. The high FPR and FNR of existing models render them impractical for real-world industrial deployments, where the cost of misidentifying anomalies is extremely high. This motivates the design of GraphNC to effectively mitigate such errors by improving the existing pre-trained GAD detectors. As the first plug-in module, GraphNC can consistently enhance detector performance—even when the teacher model performs poorly—which makes it highly practical for real-world deployment.
> > >
> > > Both above contributions are supported by a thorough theoretical analysis and extensive empirical validation in our paper. We therefore believe that this orthogonal direction is promising and should not be considered lacking in novelty, as previously suggested.
> > >
> > > We hope that the above explanations clearly demonstrate the novelty and significance of our contributions. We would be pleased to address any specific concerns or questions the reviewer may have, as we believe a more detailed and concrete discussion would be more constructive than the broad and abstract characterization of “lack of novelty.”
> > >
> > > - [Ref1] Qiao, Hezhe, et al. "Deep graph anomaly detection: A survey and new perspectives." IEEE Transactions on Knowledge and Data Engineering, 2025.
> > > - [Ref2] Ma, Xiaoxiao, et al. "A comprehensive survey on graph anomaly detection with deep learning." IEEE Transactions on Knowledge and Data Engineering, 2021
> > > - [Ref3] Liu, Kay, et al. "Bond: Benchmarking unsupervised outlier node detection on static attributed graphs." Advances in Neural Information Processing Systems，2022

---

### Official Review · Reviewer_tinT · 2025-11-01

**Soundness:** 2
**Presentation:** 4
**Contribution:** 3
**Rating:** 4
**Confidence:** 4

**Summary:**

This paper studies semi-supervised graph node anomaly detection (GAD) problem, where the model is trained with a subset of labeled normal nodes and then infers on the full graph. They propose a framework to enhance existing semi-supervised GAD methods with two components: (1) Anomaly Score Alignment, which aligns the anomaly score distirbution of the teacher model (namely one off-the-shelf semi-supervised GAD method) with those of the student model (namely the enhanced model); By using aligning with raw anomaly scores from the teacher model, the student model can better separate normal and abnormal instances in their anomaly score distribution. (2) However, the teacher model can make mistakes. In other words, the anomaly scores produced by the teacher model may be misleading. To mitigate this, they employ a regularizer to make the represenations of normal instances more compact in the latent space. One can see the idea is very intuitive but still interesting. They perform many experiments to evalute the effectiveness of their framework. Though it is a good paper already, I have two major concerns (see weak points), which should be addressed to fulfil the (high) publication standard  of ICLR.

**Strengths:**

1. This paper is very well presented, including the organisation and writing.
2. The idea is intuitive but interesting. (I will not call an intuitive idea ``lack of novelty").
3. Most claims are supported either by theoretical analysis or empirical studies.

**Weaknesses:**

##Major Concerns

1. [Implicit Assumption]: During the construction of GraphNC, the authors (implicitly) assume the teach model performs (reasonably) well. However, this is not always the case. I would like to see a study where the teacher model performs poorly (say with ROC < 0.5). In this case, most predited labels given by the teacher model are wrong (namely the anomaly scores given by the teacher model are misleading), will your framework amplify such wrong signals (and thus get worse results, i.e., lower accuracy)?

2. [Unjustified results]: In figure 4(e) and (f), one can clearly see that the performance of GraphNC first increases but then decreases with the increase of training size. This is counter-intuitive but the authors did not investigate the reasons. Instead, they wrote that ``Similar with other competing methods, with the increase of training size $R$, our model GraphNC generally performs better...", which is not true and not responsible. Importantly, one can see that GraphNC obtains the best results when the ratio is 0.15 (which corresponds to the setting of results reported in Table 1 and other main tables). However, if the ratio changes (either increases or decreases), the AUROC may substantially decreases (e.g., from 0.76 to 0.68 on Photo; Similar trends may also hold for other datasets, and I would like the authors to provide full results on all datasets regarding this.) Overall, this is a critical issue that needs to be investigated and solved. Otherwise, the conclusions given by this paper should be treated with caution.

##Minor Concerns

1. [False y-axis labels]: In Figure 1, the range of  false positive rate (and false negative rate) should be [0,1]. That is why we call it a "rate".

2. [Writing]: Page 6, ``...Note that some unsupervised methods, such as CoLA (Liu et al., 2021b), GRADATE (Duan et al., 2023), and HUGE (Pan et al., 2025), cannot be adapted to semi-supervised setting due to their design more relying on fully unlabeled data, which are excluded from our comparison." this sentence should be put behind the Competing Methods rather than Datasets.

3. [Writing]: Page 7, ``...The AUROC and AUPRC results are shown in Table 3....", it should be Table 2.

I will raise my ratings if my concerns are (partially) addresses.

**Questions:**

Please check the weak points.

---

> ### Author Response · Authors · 2025-11-20
>
> We sincerely appreciate your constructive and positive comments on our paper’s intuition and experimental analysis. Please see our response to your comments one by one below.
>
> >**Weakness #1:** . Will GraphNC amplify wrong signals when the teacher model performs poorly (say with ROC < 0.5).
>
> Thank you very much for your comments. Following your suggestion, to evaluate the performance of GraphNC when the teacher performs poorly (AUROC < 0.5), we inject anomalies into the labeled normal nodes to deliberately degrade the teacher’s performance.  When the teacher model performs poorly, we still apply score distribution alignment and NormReg to achieve normality calibration. The experimental results using GGAD as teacher model are shown in Table A1 and A2.
>
> ```
> Table A1. AUROC results of GraphNC and GGAD.
> ```
> |  Data   | Reddit | Tolokers  | Photo  | Amazon | T-Finance | YelpChi |
> |---------|--------|-----------|--------|--------|-----------|---------|
> |  GGAD   | 0.4483 |  0.4504   | 0.4465 | 0.6336 |  0.5132   | 0.5130  |
> | GraphNC | 0.5303 |  0.6291   | 0.5970 | 0.7769 |  0.7450   | 0.5685  |
>
> ```
> Table A2. AUPRC results of GraphNC and GGAD.
> ```
> |  Data   | Reddit | Tolokers  | Photo  | Amazon | T-Finance | YelpChi |
> |---------|--------|-----------|--------|--------|-----------|---------|
> |  GGAD   | 0.0276 |  0.1946   | 0.0856 | 0.1171 |  0.0443   | 0.1603  |
> | GraphNC | 0.0375 |  0.2930   | 0.1145 | 0.2274 |  0.0930   | 0.1823  |
>
>
> From the results, we observe that when the teacher's performance is poor, e.g., the AUROC is lower than 0.5 on some datasets, GraphNC can largely enhance the performance. The main reason is that the errors introduced by teacher distillation mainly affect a small portion of nodes, for example, some normal nodes may receive slightly higher scores (while most normal nodes still remain low), and some anomalies may obtain lower scores. However, NormReg relies on the explicitly labeled normal nodes, rather than the teacher’s predictions, to enforce the compactness of the normality cluster. Consequently, during the ScoreDA process, the normal cluster continues to be tightened, while the correctly predicted normal nodes guide the adjustment of other normal nodes. This helps correct some of the teacher’s mistakes, ensuring that GraphNC can still achieve performance gains even when the teacher performs poorly.
>
> Moreover, as shown in Table 2 of the paper, we observe that DOMINANT and OCGNN perform poorly on certain datasets. For instance, on Photo, DOMINANT exhibits notably low performance, suggesting that the teacher’s label distribution becomes unreliable during score alignment. GraphNC leverages NormReg to prevent overfitting to these inaccurate supervisory signals and to better exploit the limited yet accurate normal supervision, thereby improving overall performance. Finally, we summarize the average improvement of GraphNC over the teacher model across different teacher score ranges, as presented in Table A3.
>
> ```
> Table A3. Statistics of the average AUROC improvement of GraphNC w.r.t. different accuracy levels of supervision from the teacher model GGAD
> ```
> |     GraphNC   | 0.1345  | 0.1128  | 0.0764  | 0.0204
> |-----------------|---------|---------|---------|---------|
> | Teacher (GGAD) |$(0.4\sim0.5)$|$(0.5\sim0.6)$ |$(0.6\sim0.7)$ |$(0.7\sim0.9)$
>
> From the results, we observe that when the teacher performs poorly, GraphNC achieves substantially larger improvements (e.g., $\Delta (0.4\sim0.5) > \Delta (0.7\sim0.9)$. This is because the weaker teacher leaves more room for enhancement, and GraphNC is robust to different levels of noisy supervision from the teacher model. This further demonstrates the robustness and effectiveness of GraphNC in normality calibration, even when the teacher performs poorly.

---

> ### Author Response · Authors · 2025-11-20
>
> >**Weakness #2:**  Clarification on the performance of GraphNC first increases but then decreases with the increase of training size
>
> Thank you very much for pointing this out. We would like to clarify that the improvement is not strictly monotonic—in some cases, the model achieves its best performance at a particular sampling ratio. This occurs because, at certain ratios, the randomly selected normal samples happen to be of higher quality, enabling the model to better characterize normality. Given this fact,  we evaluated the full performance distribution by varying the proportion of training normal samples from 3\% to 30\% on the Reddit, Photo, Tolokers, and T-Finance datasets. The results are presented in Table A4. We observe that GraphNC performs poorly when the number of normal samples is extremely limited, as such a small subset cannot reliably capture the underlying normality. As the amount of training on normal data increases, the performance generally improves.
>
> ```
> Table A4. The AUROC results with varying the training size
> ```
> |  Data    |  3\% |  5\% |  7\% | 10\% | 15\% | 20\% | 25\% | 30\% |
> |----------|------|------|------|------|------|------|------|------|
> |  Photo   | 0.65 | 0.69 | 0.69 | 0.68 | 0.76 | 0.69 | 0.68 | 0.78 |
> |  Reddit  | 0.56 | 0.58 | 0.62 | 0.63 | 0.64 | 0.65 | 0.65 | 0.63 |
> | Tolokers | 0.53 | 0.53 | 0.55 | 0.61 | 0.65 | 0.68 | 0.66 | 0.64 |
> | T-Finance| 0.82 | 0.83 | 0.83 | 0.83 | 0.83 | 0.84 | 0.84 | 0.84 |
>
> Importantly, this fluctuation does not affect our overall conclusion: in a broader sense, GraphNC’s performance increases as the training set grows, especially when compared with settings where only a very small number of normal samples are available. We will clarify this to prevent any potential misunderstanding. Besides, it is worth noting that all baseline methods use the same default training ratio (15\%) and training data, ensuring that the comparisons are fair across all models. Moreover, as shown in Figure 4 (e) and (f) in the main paper, GraphNC consistently outperforms all baseline methods across different training sizes. We will include the remaining results and provide the corresponding figures for all datasets, and we will update the related claims accordingly in the final version of the paper.

---

> > ### Author Response · Authors · 2025-11-27
> >
> > Dear Reviewer tinT,
> >
> > We hope this message finds you well. First, we would like to express our sincere gratitude for taking the time to carefully review our work and for providing frank and insightful comments. Your feedback is extremely valuable in helping us better understand and improve our manuscript.
> >
> > In response to your review, we have made a series of substantial revisions and additions during the current discussion phase, and we have provided detailed clarifications and new experimental results above. We sincerely hope that these additional efforts have addressed your main concerns regarding the methodology of this work.
> >
> > As the discussion phase is drawing to a close, if any parts of our responses remain insufficient or unclear, we would be very grateful if you could continue to point them out. We will do our utmost to further refine and supplement the manuscript within the remaining time of the discussion.  Once again, we sincerely thank you for the time and effort you have devoted to our submission, as well as for your rigorous evaluation and constructive suggestions on our work.
> >
> > Sincerely,
> >
> > The Authors of Submission19076

---

> > > ### Comment · Reviewer_tinT · 2025-11-27
> > >
> > > Dear authors,
> > >
> > > I have carefully read your responses to my two major concerns, which are well addressed (espeically the first one). Therefore, I'd like to raise my overall rating from 4 to 6 (I did not give a score of 8 because the overall contributions of this paper are not that significant.) I hope you can incoporate my major concerns (and also how you addressed them) into the revised manuscript.

---

> > > > ### Author Response · Authors · 2025-11-27
> > > >
> > > > Dear Reviewer tinT,
> > > >
> > > > Thank you very much for your thoughtful consideration of our responses and for raising the rating to an acceptance score. We are delighted that our clarifications have satisfactorily addressed your concerns. We will carefully take your suggestions into account and incorporate the corresponding revisions into the updated version.
> > > >
> > > > Please feel free to let us know if you have any further questions or feedback. Thank you again for your valuable time and support.

---

### Official Review · Reviewer_yumf · 2025-11-09

**Soundness:** 2
**Presentation:** 3
**Contribution:** 3
**Rating:** 4
**Confidence:** 4

**Summary:**

The authors propose a framework, GraphNC (Graph Normality Calibration), designed to improve semi-supervised graph anomaly detection (GAD). The paper addresses the problem of existing models overfitting to the limited patterns of labeled normal nodes, which leads to high false positive rates. The main idea is to use a teacher-student framework to "calibrate" the normality learned by a pre-trained GAD model (the teacher). The authors propose two main components: (i) Anomaly Score Distribution Alignment (ScoreDA), which trains the student model by aligning its output scores with the teacher's score distribution across all nodes , and (ii) Perturbation-based Normality Regularization (NormReg), which mitigates the impact of the teacher's inaccurate scores by enforcing a consistency loss on augmented, labeled normal nodes to learn a more compact normal representation. The authors then demonstrate that GraphNC can be flexibly applied to different types of teacher models (e.g., reconstruction, one-class) and improves their performance across six benchmark datasets.

**Strengths:**

There are a few things I like about the paper:
1. The paper addresses a practical challenge in semi-supervised GAD: overfitting to the seen normal patterns and the resulting high false positive rate, which indicates the need for further calibration on the prediction.
2. The proposed GraphNC framework can serve as a plug-and-play module to enhance existing semi-supervised GAD models.
3. The proposed components are well-motivated. ScoreDA leverages the teacher's general knowledge, while NormReg specifically targets the teacher's weaknesses (inaccurate scores) by introducing a regularization term on labeled data.
4. The authors provide a theoretical analysis of the proposed approach.
5. The demonstrated consistent improvement over semi-supervised models without normality calibration over multiple datasets, as well as provided ablation study of the model.

**Weaknesses:**

1. [Baselines]. Some of the baselines models used as the teacher models in the paper are relatively old, like DOMINANT. I would suggest the authors add more recent baselines.
2. [Ablation]. The ablation study is insightful but reveals a potentially negative interaction. The variants with just NormReg perform poorly, sometimes even worse than the original teacher (OT). This suggests NormReg is not a standalone regularizer and is highly dependent on the ScoreDA. A deeper analysis of why it fails on its own would strengthen the paper.
3. [Motivation]. The paper started with the motivation for high FPR in the introduction. However, there is not much discussion on FPR rate in the experiments.
4. [Overhead] The framework requires pre-training a teacher model and then training a separate student model. However, the paper does not discuss the total computational overhead (training and inference time) compared to just using the (already strong) teacher model.

**Questions:**

Please address the weaknesses mentioned above.

---

> ### Author Response · Authors · 2025-11-20
>
> We sincerely appreciate your constructive and positive comments on our methodology design and empirical justification. Please see our response to your comments one by one below.
>
> >**Weakness #1:** . More recent baselines should be involved in the baselines.
>
> Thank you very much for your suggestion. Following your suggestion, we have included two additional baselines published in 2024, GADNR [Ref1] and ADA-GAD [Ref2]. The comparison results are shown in Tables A1 and A2.
>
> ```
> Table A1. AUROC comparison with additional baseline methods.
> ```
> |  Data   | Amazon | T-Finance | Reddit | YelpChi | Tolokers | Photo  |
> |---------|--------|-----------|--------|---------|----------|--------|
> |  GADNR  | 0.6436 |  0.5943   | 0.5978 | 0.4554  |  0.6406  | 0.6158 |
> | ADA-GAD | 0.2034 |  0.4416   | 0.5765 | 0.4400  |  0.5369  | 0.4067 |
> |GraphNC|**0.9613**| **0.8340** |**0.6420**|**0.6630**| **0.6505** |**0.7693**|
>
> ```
> Table A2. AUPRC comparison with additional baseline methods.
> ```
> |  Data   | Amazon | T-Finance | Reddit | YelpChi | Tolokers | Photo  |
> |---------|--------|-----------|--------|---------|----------|--------|
> |  GADNR  | 0.0866 |  0.0523   | 0.0456 | 0.1324  |  0.3511  | 0.1430 |
> | ADA-GAD | 0.0388 |  0.0375   | 0.0417 | 0.1295  |  0.2566  | 0.0723 |
> |GraphNC |**0.8403**| **0.3667**  |**0.0560**|**0.2389** | **0.3082** |**0.3561**|
>
>
> From the results, we can observe that GraphNC outperforms the two additional baselines across all benchmark datasets on both the AUROC and AUPRC. Besides, we would like to highlight that the strong baseline, GGAD [Ref3], is a very recent  method published in NeurIPS 2024.
>
> In addition, DOMINANT is a simple yet highly effective data-reconstruction-based method, and many recent reconstruction approaches, including GAN-NR and ADA-GAD, can be viewed as its variants. For this reason, we select DOMINANT as the representative model to serve as one of the teacher models.
>
> - [Ref1] Roy, Amit, et al. "Gad-nr: Graph anomaly detection via neighborhood reconstruction." WSDM, 2024.
> - [Ref2] He, Junwei, et al. "Ada-gad: Anomaly-denoised autoencoders for graph anomaly detection." AAAI, 2024.
> - [Ref3] Qiao, Hezhe, et al. "Generative semi-supervised graph anomaly detection." NeurIPS, 2024.
>
> >**Weakness #2:**  NormReg performs poorly, sometimes even worse than the original teacher.
>
> Thank you very much for your comments. We agree that in our GraphNC framework, its two components—NormReg and ScoreDA—need to work collaboratively to achieve the best performance; the individual ScoreDA or NormReg contributes to the overall performance in a very different way, with ScoreDA designed to learn from the teacher model and NormReg designed to regularize the graph normality for effectively mitigating the negative impact of inaccurate information in ScoreDA. Thus, we would like to clarify that the full approach GraphNC, rather than its individual component, is a standalone framework that can be plugged into different existing semi-supervised GAD methods to enhance their performance.
>
> In the ablation study, we observe that directly applying NormReg to the teacher model (OT) can lead to performance degradation on certain datasets. This is because NormReg is specifically designed to calibrate normality during the student’s learning of the teacher’s score distribution in ScoreDA; it is not designed to work as a standalone component. The results of OT + NormReg in the ablation study are presented to highlight the results of OT + ScoreDA + NormReg, thereby justifying our insight into the synergy between ScoreDA and NormReg. We will clarify this point in our final version.
>
> >**Weakness #3:** More analysis on the FPR rate.
>
> We appreciate your suggestion. Note that according to Figures 1 (a) and (b), we provided the discussion on the FPR and TPR in the introduction section. In particular, due to the large overlapped anomaly scores for the normal and abnormal nodes,  existing GAD methods suffer from high detection errors, e.g., normal nodes that are dissimilar to the labeled normal nodes are detected as anomalies (false positives) and vice versa (false negatives). In the main experiments,
> we report both AUROC and AUPRC, which are two widely used metrics for anomaly detection, where the AUPRC results also highlight the effectiveness of GraphNC, since lower false-positive rates lead to higher precision and therefore a higher AUPRC. Following your suggestion, we will provide the FPR values for the remaining datasets and add more discussion on the FPR rate in the experimental analysis in the final version of the paper.

---

> > ### Author Response · Authors · 2025-11-20
> >
> > >**Weakness #4:** The total computational overhead (training and inference time) compared to just using the (already strong) teacher model.
> >
> > Thank you so much for pointing out this problem. Note that we analyzed the time complexity of the proposed plug-in module in Appendix E. Following your suggestions, we also provide a runtime comparison between the standalone teacher model and the model augmented with GraphNC below. The training and inference time comparison are shown in Table A3 and Table A4.  It is clear that GraphNC is highly efficient for training, as its training process builds directly upon the pretrained teacher model. In addition, GGAD and GraphNC are both very efficient for inference.
> >
> > ```
> > Table A3.  Runtimes (in seconds) of training on the six datasets of GGAD and GraphNC.
> > ```
> > |  Data   | Amazon | T-Finance | Reddit | YelpChi | Tolokers | Photo  |
> > |---------|--------|-----------|--------|---------|----------|--------|
> > |  GGAD   |  658   |   9345    |  368   |  480    |   850    |   106  |
> > | GraphNC |  60    |   3050    |  900   |  4200   |   19     |   89   |
> >
> >
> > ```
> > Table A4.  Runtimes (in seconds) of inference on the six datasets of GGAD and GraphNC.
> > ```
> > |  Data   | Amazon | T-Finance | Reddit | YelpChi | Tolokers | Photo  |
> > |---------|--------|-----------|--------|---------|----------|--------|
> > |  GGAD   |  0.0991  | 1.0896 | 0.0867 | 1.4931 | 0.1023 | 0.0508 |
> > | GraphNC |  0.0951  | 1.0929 | 0.0785 | 1.5323 | 0.0934 | 0.0403 |

---

> > > ### Author Response · Authors · 2025-11-28
> > >
> > > Dear Reviewer yumf,
> > >
> > > We hope this message finds you well. We would like to express our sincere appreciation for the time you have spent reviewing our work and for your insightful comments. Your feedback has been invaluable in helping us strengthen and refine our manuscript.
> > >
> > > In response to your review, we have made substantial revisions and additions during the discussion phase, accompanied by detailed clarifications and new experimental results. We sincerely hope that these efforts have addressed your primary concerns regarding the methodology of this work.
> > >
> > > As the discussion phase is drawing to a close, if any aspects of our responses remain unclear or insufficient, we would be deeply grateful if you could kindly point them out. We will do our utmost to further improve and supplement the manuscript within the remaining time.
> > >
> > > Once again, thank you very much for your time, effort, and constructive evaluation of our submission. Your guidance is greatly appreciated.
> > >
> > > Sincerely,
> > > The Authors of Submission19076

---

> > > > ### Comment · Reviewer_yumf · 2025-11-28
> > > >
> > > > Thank you to the authors for proving responses to my concerns.
> > > > It addressed some of my concerns, though, I would prefer to see more baselines comparison with recent methods.
> > > > I will update the score in the next phase of review process.

---

> > > > > ### Author Response · Authors · 2025-11-28
> > > > >
> > > > > Dear Reviewer yumf,
> > > > >
> > > > > Thanks for the follow-up comments. We have compared our method with many (seven in the paper + two in the rebuttal) recent state-of-the-art GAD methods, including both semi-supervised and adapted unsupervised methods from different categories of approaches. It is true that there are always more baseline methods we could include, but we have compared with the most relevant ones to justify the claims we made. We would appreciate if you could point out which missing baselines you find relevant and want to see the comparison to these methods. We could then directly respond to your concerns.
> > > > >
> > > > > In addition to this baseline concern, may we check whether there are other remaining concerns?
> > > > >
> > > > > Thank you very much again for your help in enhancing our work!

---

### Meta-Review · Area_Chair_X9xm · 2025-12-30

**Summary:**

Two reviewers are supportive, and the others are negative. The major concern of the negative reviewers is that the novelty of the proposed method is not significant, as it is based on several existing approaches. The authors made the following explanation for the novelty and significance:
1) The method is tailored to a specific problem of anomaly detection, which hasn't been addressed before.
2) The paper provided some theoretical results to justify the effectiveness of the method.

I spent a few hours reading the paper and agree with the rebuttal of the authors. I think the merit of the work is that:
1) It used simple approaches to address a challenging problem.
2) The numerical results are very impressive and promising.

I think that this paper is borderline, but should be rejected if we consider that there are many papers with higher novelty and more complete experiments.

I suggest that the authors consider the following modification in the next version of the paper:
1) Use a list of items to summarize the contribution of each block of the proposed model to the overall performance.
2) Include all of the experiments produced during or after the rebuttal in the revised paper.
3) Enhance the proof for Theorem 1. For instance, the statement "the variance of the predicted scores of the student model on normal nodes is usually not larger than that of the teacher model" is not rigorous.

**Reviewer Concerns:**

**Reviewer yumf**
1. The baselines in the experiments are insufficient.
2. The analysis for the ablation studies is insufficient.
3. The discussion on FPR in the experiments is insufficient.
4. The total computational overhead hasn't been discussed.

I think that concerns 1, 2, 3, and 4 have been addressed by the rebuttal, in which the authors added more experiments. The reviewer was willing to update the score.

**Reviewer tinT**
1. The dependence on the accuracy of the teacher model hasn't been clearly analyzed.
2. The counter-intuitive results, e.g. those in Figure 4(e) and (f) haven't been clearly explained.

I think that the two concerns have been addressed. In fact, during the rebuttal, the reviewer stated that the concerns had been addressed.

**Reviewer XpiH**
1. The novelty of the paper is limited.
2. The method assumes the teacher produces reasonably accurate scores for many nodes, which may not always hold.

The authors provided some explanation regarding the two concerns, which, I think, have been partially addressed.

**Reviewer GNkY**
1. The novelty of the proposed method is insufficient, due to the straightforward combination of existing ideas.
2. The paper hasn't fully explained where the actual performance gain comes from.
3. There exists a conceptual inconsistency with the teacher–student paradigm and semi-supervised learning.

I think that the rebuttal has addressed concerns 2 and 3, while concern 1 remains to some extent.

**Reviewer Scores:**

Reviewer yumf would raise the score from 4 to 6, since he/she mentioned that some of the concerns had been addressed and he/she was willing to update the score. Reviewer tinT would raise the score to 6, as he/she already mentioned this during the rebuttal (before the Openreview bug). I think it is quite difficult for Reviewer XpiH and Reviewer tinT to raise their scores, since their major concerns are about the novelty of the work. They would keep their negative scores, i.e., 4 and 2 (or 4 at most). Therefore, the expected final average score remains below 5.

---

### Decision · Program_Chairs · 2026-01-26

Reject